# ENERGY CALIBRATION HEAD: A PLUG-IN NEURAL NETWORK HEAD WITH HUMAN-LIKE UNCERTAINTY

## ABSTRACT

The ability to distinguish what it knows from what it does not, known as metacognition, has been one of the fundamental challenges in modern AI. One benefit of metacognition is that it could preclude overconfident learning about out-of-distributions. For instance, machine learning models often exhibit excessive confidence when dealing with uncertain inputs. To mitigate this issue, we leverage the relationship between the marginal probability and conditional uncertainty found in our human behavioral experiments classifying out-of-distribution (OOD) images. Theoretical analyses reveal that uncertainty and marginal energy are loosely related and significantly influenced by the latent vector norm. Building upon this finding, we propose a novel plug-in type layer: energy calibration head (ECH). The ECH uses a metacognition module that calibrates uncertainty by evaluating the difference between actual marginal energy (indicative of how much it knows) and the marginal energy predicted based on the uncertainty level, leading to the attenuated joint energies for the OOD samples. We showed that a neural network with ECH emulates human-like uncertainty in OOD images (45.1% AUROC error reduction on average compared to a linear head) and can effectively perform anomaly detection tasks.

## 1 INTRODUCTION

Despite the recent astounding advancements in machine learning, the challenge of imbuing machines with metacognition [16, 15], the ability to discern what they know from what they do not, remains an open question. It is widely acknowledged that while humans and animals identify uncertain input seemingly effortlessly [36, 34, 15], machines often become overconfident when encountering unseen samples [56, 29].

This study delves into this overconfidence issue within the domain of uncertainty, focusing on out-of-distribution (OOD) samples [40, 28, 29]. Specifically, it has not been fully understood **i**) why overconfidence occurs in neural classifiers and **ii**) how to establish a human-like uncertainty calibration mechanism by metacognition. While previous works [8, 30, 75] have made significant advances in OOD detection and generalization, recent studies utilize human annotators' behavioral data to effectively reduce the gap between human and machine metacognitive capacities [58, 63, 7].

Given human metacognition, we reasoned that the probability of the chosen decision being correct should closely track their actual performances in the noisy inputs. In line with this intuition, previous studies on the human classification tasks on OOD samples showed that not only are the strongly perturbed images perceived as unlikely 'noise' [78], resulting in the accuracy drops [20], but also the predicted class labels vary among human subjects [9]. In our study, we collected the participants' population-level decision probability and class conditional uncertainty through online experiments. We got a hint about the intriguing connection between marginal density and conditional uncertainty. Based on these empirical observations, our theoretical analyses further uncover the interplay between tractable marginal energy and conditional uncertainty by conceptualizing generic classifiers as energy-based models [22]. Specifically, we found that the latent norm directly influences conditional uncertainty and marginal energy, potentially disrupting the intrinsic connection. This adverse effect becomes significant when dealing with OOD samples associated with high marginal energy and conditional uncertainty. In addition, we demonstrate that samples with different marginal energies can have the same level of conditional uncertainty.

To circumvent such limitations of machine-learning models, we propose a plug-in output layer named Energy Calibration Head (ECH). The ECH can be trained to rectify the inconsistency in the relationship between uncertainty and marginal energy, thereby mitigating overconfidence without additional training or data. Specifically, our approach involves several key components. First, we eliminate the norm of latent vectors and biases, the two factors identified as potential causes of adverse effects in our theoretical analysis. Second, we add a simple meta-cognition module incorporating a compact and trainable multi-layer perceptron, which aims to predict marginal energy based on the maximum log-likelihood and expected logits as inputs. Subsequently, the energy is adjusted using a scaling factor that is inversely proportional to the difference between the predicted and actual marginal energy obtained from logits. We demonstrate that this scaling factor effectively calibrates uncertainty and resolves the overconfidence issue from both theoretical and empirical perspectives.

Experimental analyses comparing ECH with various heads showed that ECH has good uncertainty calibration performance; uncertainty increases proportionally with the distance from the in-distribution (ID) sample. When the behavior of the ECH was compared with human confidence using the CIFAR-10H [58, 2], a dataset that captures human response probabilities, we found that the ECH's responses align more closely with human behavior than those produced by the other types of heads (45.1% AUROC error reduction on average compared to a linear head). Furthermore, we evaluated the model's response to anomalies, including adversarial [49] and OOD samples. Notably, the ECH demonstrated superior performance in adversarial and OOD detection tasks. Specifically, it achieved stable performance in OOD detection, showing a decrease of 7.09% in the False Positive Rate (FPR) and an increase of 1.62% in the Area Under the Curve (AUC) compared to the best baseline for each measure.

The contributions of our work are as follows:

- **Human Behavioral Study**: We establish a link between the uncertainty in human perceptual decision-making and the marginal probability (Sec. 2).
- **Neural Model Analysis**: By relating generic classifiers to energy-based models, we examine the interplay between the conditional uncertainty and the marginal energy that often leads to an overconfident decision (Sec. 3).
- **Energy Calibration Head**: To address the overconfidence issue, we propose a novel plug-in type output head to resolve the uncertainty-marginal energy inconsistency, offering several advantages: capability of in-network calibration, adaptability owing to its plug-in nature, and no requirement of additional training or data (Sec. 4).
- **Empirical Validation**: Comprehensive analyses and simulations demonstrate the effectiveness of the ECH in uncertainty calibration and its applicability in anomaly detection. A neural network with ECH emulates human-like uncertainty in OOD images with 45.1% AUROC error reduction compared to conventional models (Secs. 5 and 6).

## 2 Marginal Energy and Conditional Uncertainty Are Coupled in Humans

To measure the conditional uncertainty for OOD samples from human behavior, we conducted online behavioral experiments using CloudResearch [45], where 250 participants were recruited to classify ID and OOD sample images $\mathbf{x}'$s using CIFAR-10 [38] dataset $\mathcal{D}$. Ten sets were created, each comprising 125 images, and presented to 25 participants. Images were generated by adding random noise on the corresponding ID samples: $\mathbf{x}' = (1 - \lambda)\mathbf{x}_i + (\lambda)\mathbf{n}$, where $\mathbf{x}_i \in \mathcal{D}$, $\mathbf{n} \sim U(0, 1)^{W \times H \times D}$, and $\lambda \in [0, 0.9]$. The $\lambda$ represents the distance from the ID sample's location, and $\lambda = 1$ represents a complete noise (Details of the experiment, including the selection of lambda values, are in Appendix D). Each participant (compensated by USD 1.5) classified 125 images into the best-matching categories.

Consistent with the previous studies focusing on the accuracy profiles [9, 78, 21, 20, 14], we found the classification accuracy decrease for strongly perturbed images (Fig. 1a). Additionally, we analyzed the proportion of subjects who chose the most frequently chosen classes $y'$, which corresponds to $\max_{y'}(p(y'|\mathbf{x}))$. As predicted, these conditional probabilities decrease as $\lambda$ increase (Fig. 1b). Moreover, when we computed the entropy of the decision probabilities, we found that the entropy increases as a function of $\lambda$, indicating that the chosen class labels gradually diversify. These findings collectively suggest that conditional probabilities over categories converge to similar

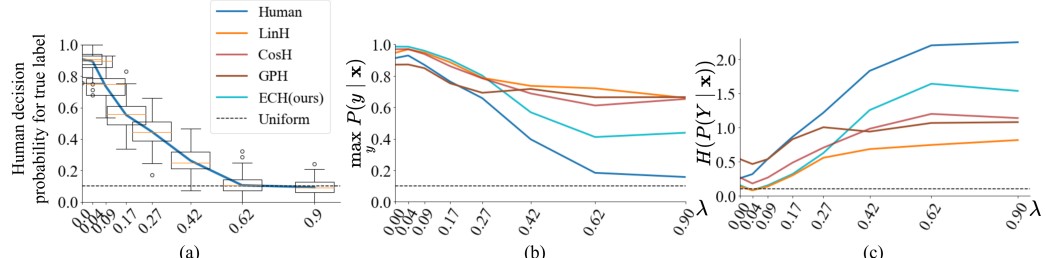

Figure 1: Exploring decision probability and uncertainty in human and model decision-making across diverse data scenarios; horizontal axis: $\lambda$ value. ID $\rightarrow$ OOD samples as $\lambda \rightarrow 1$. **(a)** The population-level human decision probability for true label samples. Line (mean probability) and box plots represent across subjects $P(y|\mathbf{x})$. **(b)** $\max_{y'}(p(y'|\mathbf{x}))$ and **(c)** entropy $H(P(y|\mathbf{x}))$ of the class conditional likelihood of human and machine classifiers with various heads: LinH, CosH, GPH, and ECH (see Appendix E for details). The proposed plug-in module (ECH) demonstrates its ability to closely simulate human behavior, particularly for out-of-distribution (OOD) samples, at higher $\lambda$ values.

levels, making it difficult to make confident decisions for the OOD samples (Fig. 1c). This behavior aligns with our intuition that the marginal density decreases as a function of $\lambda$, increasing conditional uncertainty in human decisions. Taken together, our experiments reveal that marginal energy and conditional uncertainty depend on each other in human perceptual decision-making. This also suggests the possibility that humans' uncertainty profile is suitable for recognizing OOD samples without additional training or data.

## 3 MARGINAL ENERGY AND CONDITIONAL UNCERTAINTY ARE LOOSELY RELATED IN MODELS

Fig. 1 (b-c) examines the maximum likelihood and conditional uncertainty for noisy inputs across various model heads. Unlike humans, the models exhibit difficulty in effectively calibrating uncertainty for inputs with low marginal probability. This section focuses on a theoretical and empirical investigation to understand the underlying causes of overconfidence, especially in the linear head.

### 3.1 MARGINAL ENERGY PERSPECTIVE

This section attempts to investigate the classifier's attributes within the context of an energy-based model framework, as in [47]. Specifically, we introduce the connection between tractable marginal energy and conditional uncertainty, adopting the view that a classifier can be interpreted as the energy-based model [22].

Consider a supervised setting involving an input variable $X$ and a label variable $Y$ with a classifier $f : \mathcal{X} \rightarrow \mathbb{R}^{|\mathcal{Y}|}$, where $f$ maps from input space $\mathcal{X}$ to $|\mathcal{Y}|$-dimensional logit vector space. Let $\mathcal{D} = \{(\mathbf{x}, y)_i\}_{i=1}^{N}$ denote the dataset, where $\mathbf{x} \in \mathcal{X} \subseteq \mathbb{R}^d$ represents input and $y \in \mathcal{Y}$ is the corresponding label. Then, by using the softmax function, the vector is projected to probability simplex and interpreted as conditional likelihood, *i.e.*, $P(y|\mathbf{x}) = \frac{\exp(f_y(\mathbf{x}))}{\sum_{y'} \exp(f_{y'}(\mathbf{x}))}$, where $f_y(\mathbf{x})$ means $y$-th element of logit vector $f_\theta(\mathbf{x})$. This provides a joint distribution of $(\mathbf{x}, y)$ with this logits $p(\mathbf{x}, y) = \frac{\exp(f_y(\mathbf{x}))}{Z}$, where $Z = \int_{\mathbf{x}'} \sum_{y'} \exp(f_{y'}(\mathbf{x}'))$ is an intractable normalizing constant. In this context, a logit represents the negative joint energy, *i.e.*, $f_y(\mathbf{x}) = -E(\mathbf{x}, y)$. The marginal probability $p(\mathbf{x}) = \sum_{y'} p(\mathbf{x}, y') = \frac{\sum_{y'} \exp(f_{y'}(\mathbf{x}))}{Z}$ can be obtained by marginalizing out the label $y$ from the joint probability. The marginal energy is represented as

$$E(\mathbf{x}) = -\log \sum_{y'} \exp(f_{y'}(\mathbf{x})). \tag{1}$$

This way, a generic classifier can be considered as an energy-based model that encodes the information of $p(\mathbf{x})$.

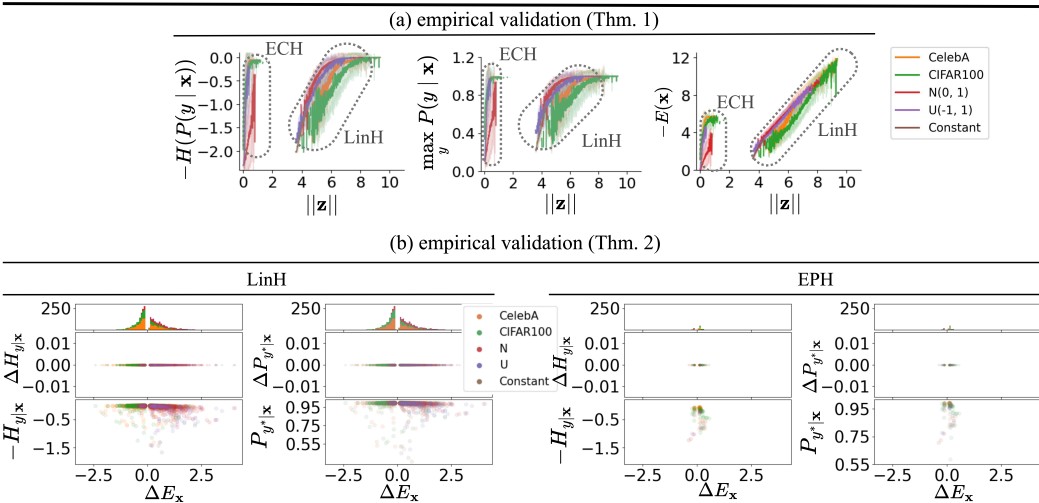

Figure 2: Empirical validation of Thms. 1 and 2 was conducted using the models with LinH and ECH trained on ID (CIFAR-10) samples. Datasets that have not been used to assess OOD detection are selected; ID: CIFAR-10 [38], OOD: CIFAR-100 [38], CelebA [48], Gaussian Noise $\mathcal{N}(0,1)$, Uniform Noise $U(-1,1)$, Constant (See Appendix F.4 for the details). Line plot in (a): (solid line) mean value and (shaded region) 1 standard deviation. (a) shows the effect of $\|\mathbf{z}\|$ on uncertainty and marginal energy, as predicted by Thm. 1. (b) presents the observation of sample pairs corresponding to the Thm. 2, especially Lemma 2; each plot is a stack of three subplots, where the horizontal axis represents $\Delta E_{\mathbf{x}} = E(\mathbf{x}_{ID}) - E(\mathbf{x}_{OOD})$; (top) exhibits a histogram, (middle) displays (left) $\Delta H_{y|\mathbf{x}} = H(P(y|\mathbf{x}_{ID})) - H(P(y|\mathbf{x}_{OOD}))$ or (right) $\Delta P_{y^\star|\mathbf{x}} = P(y^\star|\mathbf{x}_{ID}) - P(y^\star|\mathbf{x}_{OOD})$, influencing the sample pairs selection, and (bottom) showcases $-H_{y|\mathbf{x}} = -H(P(Y|\mathbf{x}))$ and $P_{y^*|\mathbf{x}} = P(y^*|\mathbf{x})$ for the ID samples in pairs; (left two) for LinH, (right two) for ECH, validating Thm. 2.

## 3.2 IMPACT OF THE LATENT NORM AND BIASES ON UNCERTAINTY AND MARGINAL ENERGY

This section shows that **i)** the latent norm directly affects uncertainty and marginal energy and **ii)** the bias modulates how latent norm affects uncertainty.

A conventional, generic classifier employs a linear output head, represented by $f_y(\mathbf{x}) = \mathbf{w}_y^\top g(\mathbf{x}) + b_y$, where $\mathbf{w}_y$ is the weight vector of the output head for class $y$, $g : \mathcal{X} \to \mathcal{Z}$ is an encoding function mapping from the input space to the latent space, and $b_y$ is the bias for class $y$. Let $\mathbf{z} = g(\mathbf{x})$ be the latent vector corresponding to the input. Then the logit can be rewritten as $f_y(\mathbf{x}) = \mathbf{w}_y^\top \mathbf{z} + b_y = \|\mathbf{z}\|\|\mathbf{w}_y\|\cos(\theta_{(\mathbf{w}_y, \mathbf{z})}) + b_y$, where $\cos(\theta_{(\mathbf{w}_y, \mathbf{z})})$ represents the cosine similarity between $\mathbf{w}_y$ and $\mathbf{z}$. For notational simplicity, we describe the logit as follows:

$$f_y(\mathbf{x}) = a(\mathbf{x}) \cdot h_y(\mathbf{x}) + b_y,$$

where $a(\mathbf{x})$ is a scaling factor independent of label and $h_y(\mathbf{x})$ is the $y$-th pre-logit depending on a label. The latent norm $\|\mathbf{z}\|$ functions as a scaling factor and $\|\mathbf{w}_y\|\cos(\theta_{(\mathbf{w}_y, \mathbf{z})})$ as a pre-logit. Once the logit is obtained from the model, we can compute the conditional likelihood as $P(y|\mathbf{x}) = \frac{\exp(f_y(\mathbf{x}))}{\sum_{y'} \exp(f_{y'}(\mathbf{x}))}$ and the entropy of conditional likelihood as $H(P(Y|\mathbf{x})) = H(\{P(y|\mathbf{x})\}_{y \in \mathcal{Y}}) = -\sum_y P(y|\mathbf{x}) \log P(y|\mathbf{x})$.

The following Thm. 1 elucidates how a scaling factor can affect the uncertainty, including entropy $H(P(Y|\mathbf{x}))$ and maximum of the conditional likelihood for a given input $\max_y P(y|\mathbf{x})$, and marginal energy $E(\mathbf{x})$ (refer to Appendix A for proof).

**Theorem 1.** *Given* $f_y(\mathbf{x}) = a(\mathbf{x}) \cdot h_y(\mathbf{x}) + b_y$, $f_y(\mathbf{x}) \neq f_{y'}(\mathbf{x})$ *for some* $y$ *and* $y'$. *For* $0 < a(\mathbf{x}) < \infty$, *the following statements hold.*

    *(i) As* $a(\mathbf{x})$ *increases with* $h_y(\mathbf{x})$ *fixed for every* $y$, $H(P(Y|\mathbf{x}))$ *decreases if* $\mathbb{E}_{P(y|\mathbf{x})}[b_y \tilde{h}_y(\mathbf{x})] > -a(\mathbf{x})\mathrm{Var}_{P(y|\mathbf{x})}(\tilde{h}_y(\mathbf{x}))$, *where* $\tilde{h}_y(\mathbf{x}) = h_y(\mathbf{x}) - \sum_{y'} P(y'|\mathbf{x})h_{y'}(\mathbf{x})$.

*(ii) Suppose $\arg\max_y P(y|\mathbf{x})$ has a single element $y^*$. As $a(\mathbf{x})$ increases, $P(y^*|\mathbf{x})$ increases if $b_{y^*} \leq \bar{b}$, where $\bar{b} = \sum_{y'} P(y'|\mathbf{x})b_{y'}$.*

*(iii) As $a(\mathbf{x})$ increases, $E(\mathbf{x})$ decreases if $\bar{h}(\mathbf{x}) > 0$, where $\bar{h}(\mathbf{x}) = \sum_y P(y|\mathbf{x})h_y(\mathbf{x})$.*

According to Thm. 1, the latent norm directly influences both uncertainty and marginal energy, and the influence on uncertainty is closely tied to the effect of biases. These characteristics present a challenge for out-of-distribution (OOD) samples that are expected to have low marginal probability and high uncertainty. Note that the condition of the fixed $h_y(\mathbf{x})$ of a linear head of a neural network implies keeping a cosine similarity $\cos(\theta_{(\mathbf{w}_y,\mathbf{z})})$ constant while allowing variations in the latent norm. In addition, the $\arg\max$ function yields a set of arguments that correspond to the maxima as its output. If the cardinality of the $\arg\max$ for the conditional likelihood is greater than one, their likelihoods are bounded by 50%, which does not depict a situation of overconfidence. Therefore, we focus only on cases with a single maximum likelihood. The empirical evidence presented in Fig. 2a further substantiates the effect of the latent norm on uncertainty and marginal energy. Although the effect does not necessarily have to be adverse, it can limit the uncertainty calibration process.

### 3.3 WEAK RELATIONSHIP BETWEEN MARGINAL ENERGY AND CONDITIONAL UNCERTAINTY

This section extends the relationship between marginal energy and conditional uncertainty. We present Thm. 2 to show that samples associated with different magnitudes of marginal energies can have the same conditional uncertainty value (refer to Appendix A for proof).

**Theorem 2.** *There exist pairs of samples $\mathbf{x}$ and $\mathbf{x}'$ satisfying each of the following conditions:*

*(i) $E(\mathbf{x}) \neq E(\mathbf{x}')$ and $H(P(Y|\mathbf{x})) = H(P(Y|\mathbf{x}'))$.*

*(ii) $E(\mathbf{x}) \neq E(\mathbf{x}')$ and $p(y^*|\mathbf{x}) = p(y^\star|\mathbf{x}')$, where $y^*$ and $y^\star$.*

This implies that the relationship between marginal energies and conditional uncertainty in models does not match that of humans (as shown in Sec. 2). The observations in Fig. 2b confirm the view; a considerable number of sample pairs are satisfying these conditions. Our theoretical analysis and empirical valuation support the view that marginal energies and conditional uncertainties are not strongly coupled.

## 4 ENERGY CALIBRATION HEAD

Expanding on the insights from Sec. 3, we introduce a plug-in head ECH designed to rectify the inconsistency in the relationship between uncertainty and marginal energy. This output head computes energy as the product of three components: cosine similarity, the scaling factor from the metacognition module, and the scaling factor from the Kernel. These scaling factors directly calibrate uncertainty and marginal energy, as the theorem demonstrates.

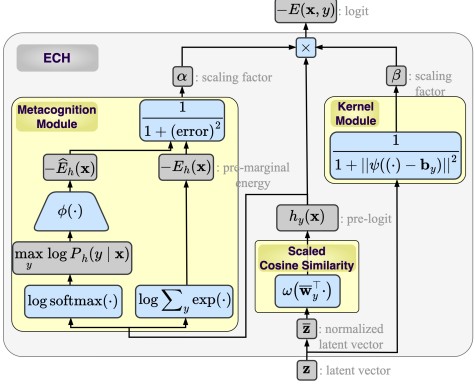

Figure 3: Schematic design of ECH

### 4.1 DESIGN OF HEAD

**Scaled Cosine Similarity**: The ECH calculates the scaled cosine similarity between the latent vector and the weight vector of the output layer, denoted as $\omega\cos(\theta_{(\mathbf{w}_y,\mathbf{z})})$, which serves as the energy component. Here, $\omega$ represents a trainable parameter akin to $\|\mathbf{w}_y\|$ in the linear head, albeit shared across classes. Consequently, we exclude the norms of latent vectors and biases from the head since these factors have been identified as potential sources of adverse effects in our theoretical analysis (refer to Thm. 1).

**Metacognition Module**: The module learns the relationship between marginal energy and uncertainty from ID samples. When a sample deviates from the learned relationship, the module rectifies

conditional uncertainty by considering the sample unseen. The rectification is accomplished by multiplying the energy and the module's scaling factor output, inducing joint and marginal energy adjustments. To distinguish the energy adjustment before and after, the marginal energy calculated according to Eq. (1) using pre-logits is referred to as pre-marginal energy $E_h(\mathbf{x})$. Likewise, the likelihood obtained from pre-logits is pre-likelihood $P_h(y|\mathbf{x})$. As shown in Fig. 3, the module use the scaled cosine similarity $\omega \cos(\theta_{(\mathbf{w}_y, \mathbf{z})})$ as the pre-logit. The module tries to check the error in between the pre-marginal energy $E_h(\mathbf{x})$ derived from Eq. (1) and the pre-marginal energy $\hat{E}_h(\mathbf{x})$ predicted using an uncertainty metric. Specifically, a three-layer perceptron $\phi$ is employed to learn a mapping from the uncertainty metric, maximum log pre-likelihood $\max_y \log P_h(y|\mathbf{x}) = \log \mathrm{softmax}(h_y(\mathbf{x}))$, to negative pre-marginal energy $-E_h(\mathbf{x})$. With an ID sample that aligns effectively with the learned mapping, the predicted pre-marginal energy will coincide with the pre-marginal energy. Conversely, a disparity arises when this relationship is inconsistent. The head rectifies the conditional uncertainty using a scaling factor $\alpha \in (0, 1]$ inversely proportional to the disparity:

$$\alpha(\mathbf{z}) = \frac{1}{1 + (\hat{E}_h(\mathbf{x}) - E_h(\mathbf{x}))^2}.$$

This process allows an uncertainty calibration based on the learned relations, enhancing the metacognitive capability of the model.

**Kernel module**: The module outputs a scaling factor that converges to zero for sufficiently faraway samples by adopting the kernel, increasing conditional uncertainty. The motivation of this module is based on distance-based OOD detection scores [43, 1, 61, 32, 76, 60, 10, 66, 69] proven to be successful in OOD detection. As depicted in Appendix H.3, distance and cosine similarity appear suitable for distinguishing between ID and OOD samples. Instead of adopting a post-hoc detection score, we use this concept to in-network uncertainty calibration. The scaling factor incorporates the distance from the learned bias in the latent space:

$$\beta(\mathbf{z}) = \frac{1}{1 + \|\psi(\mathbf{z} - \mathbf{b}_y)\|^2},$$

where $\psi : \mathbb{R}^{d_l} \rightarrow \mathbb{R}^{d_k}$ is a trainable non-linear projection function, $\mathbf{b}_y \in \mathbb{R}^{d_k}$ is a bias term expected to act as a class-wise mean, $d_l$ and $d_k$ are the dimensions of the latent and projection space, respectively. It resembles the projection head in contrastive learning [5, 23, 3, 35, 52] but with the distinction that it is defined on a class-wise projection (the kernel and projection function selection is discussed in the Appendix).

**Energy of ECH** The energy output is determined by multiplying all three factors:

$$-E(\mathbf{x}, y) = \alpha(\mathbf{z})\beta(\mathbf{z})\omega \cos(\theta_{(\mathbf{w}_y, \mathbf{z})}).$$

## 4.2 Training and Modeling

**Traning**: During training, the module is exposed to ID samples to minimize the difference. The training process relies on cross-entropy loss without introducing supplementary loss for the trainable parameters employed in an ECH. By the cross-entropy loss, we can see that the energy corresponding to the true label increases while the remaining energy decreases. This process implies that the scaling of the energy deviations is learned to increase, ultimately driving the error in the metacognition module toward zero (See Appendix H.1 for empirical evidence).

**Modeling Benefits**:

- (In-model calibration.) The ECH effectively mitigates the problem of overconfidence within the network by accurately calibrating the uncertainty of out-of-distribution (OOD) samples through the meta-cognition module by leveraging the relationship between uncertainty and marginal energy.
- (Plug-in type head.) The ECH modifies the energy calculation of the output layer while maintaining the encoder's overall structure. Since the ECH can replace only the output layer of a generic class of neural networks, it can seamlessly complement various existing neural models and learning methods, enhancing overall efficacy.
- (No additional training or data.) The manifold learning assumes that samples in a high-dimensional space are distributed over a low-dimensional subspace, implying that the data manifold's volume is considered zero. When dealing with OOD samples, fitting the model to calibrate uncertainty

becomes challenging, as it requires a high-capacity model, extensive training time, and substantial data. Therefore, we attempt to address the overconfidence of neural classifiers when confronted with OOD samples without additional training or extra data. The uncertainty calibration of ECH is accomplished without requiring additional training. Meanwhile, various studies within the domain of Out-of-Distribution (OOD) detection often resort to either an independent synthetic dataset or necessitate incorporating an auxiliary learning mechanism. Notably, our approach only introduces the simple modification of the model head. In addition, it eliminates the need for extra training data, such as unexperienced [8, 30, 75, 53, 4, 51, 33], synthesized [41, 62, 67, 12, 11], or sampled data [13, 22, 72, 73], significantly simplifying the implementation process. However, data augmentation is still employed because it is used to learn a conventional classifier.

## 5  EMPIRICAL ANALYSIS OF UNCERTAINTY CALIBRATION

We conducted a comprehensive analysis to empirically validate the ECH's effectiveness in uncertainty calibration. Compared were the models with different heads: a linear head (LinH), a cosine similarity head (CosH), a Gaussian Process head (GPH), and an ECH (See Appendix E for details of each head). The model architecture was fixed as WideResNet (WRN)-28-10 [77]. The models were trained using the CIFAR-10 training dataset as the ID data. All models were implemented using PyTorch [57], a Python package that supports automatic differentiation, and the computations were performed on NVIDIA GeForce RTX 3090 GPUs. We used established hyperparameters for training and neural architecture without engaging in extensive hyperparameter tuning. Additionally, exploratory experiments were conducted for the ECH, involving variations in module combinations, hyperparameter configurations, and backbone network architectures (See Appendix G).

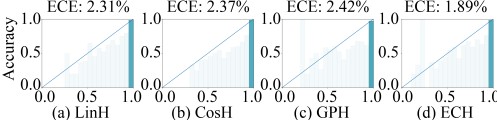

Figure 4: Calibration of accuracy vs. model confidence. The horizontal axis represents the conditional likelihood of each head, while the vertical axis indicates the human decision probability within each bin. Transparency is based on the frequency of each bin.

Figure 5: Entropy of distant samples with each legend representing a specific head. The horizontal × vertical axes denote the two-dimensional input subspace, and the depth axis signifies entropy. The columns correspond to different distance scales.

**ID Calibration**  One inherent implication of classifiers with better meta-cognitive capabilities is a stronger correlation between predicted probabilities and the actual accuracy of the classifier. In a well-calibrated model, confidence closely aligns with accuracy. We evaluated the calibration performance of the model with respect to accuracy on the test set of ID samples using the Expected Calibration Error (ECE) (refer to Appendix F.1 for details). As depicted in Fig. 4, LinH yields an ECE of $2.31\%$, CosH $2.37\%$, and GPH $2.42\%$. In contrast, ECH achieves $1.89\%$ ECE, suggesting that ECH calibrates confidence more effectively in ID datasets than the others.

**OOD Calibration**  Note that in this study, the uncertainty calibration extends beyond verifying the meta-cognitive ability of the model; it aims to affirm the consistency between the model's predictions and human predictions, pursuing human-like uncertainty. Since the true labels of ID samples for classification rely on human judgments, we regard humans as oracles. The outcomes of the human behavioral study in Sec. 2 provide a benchmark for evaluating the model's resemblance to human behavior in OOD samples. The maximum conditional likelihood and entropy of heads for the OOD samples are illustrated in Fig. 1. As demonstrated in Fig. 1a, humans' confidence in the true label diminishes and converges to $1/|\mathcal{Y}|$ as the $\lambda$ increases while moving away from the ID. Figure 1b displays the maximum conditional likelihood of humans and each head. All heads manifest similar behavior at low $\lambda$ (close to ID). However, with increasing $\lambda$ (ID → OOD), ECH gradually reduces confidence, whereas the other heads exhibit a slower decline. Comparing the AUROC of maximum likelihood from humans to the models, ECH demonstrates a relatively lower AUC Error (AUCE) 0.1644 than LinH (0.2935). Figure 1c illustrates the entropy of conditional likelihood. Humans

exhibit higher entropy toward the lower negative marginal region, and ECH aligns with this more closely than the other heads. The quantitative results are presented in Appendix H.2.

**Uncertainty Calibration for Distant Samples**    We investigated how uncertainty is calibrated for samples relatively far from the identity (ID) samples. Initially, sampling is done in two random directions for each ID sample. We generated grid inputs in a two-dimensional subspace spanning these directions. The study evaluated the conditional entropy for each given grid input $H(P(Y|\mathbf{x}))$. As shown in Fig. 5, the ECH shows a clear trend where uncertainty rapidly increases as samples move away from the ID sample. Furthermore, details of the analysis are provided in Appendix F.2. Refer to Appendix H.4 for quantitative results.

## 6    ANOMALY DETECTION

To validate the efficacy of uncertainty calibration of ECH, we conducted two anomaly detection experiments: adversarial and OOD detection.

### 6.1    ADVERSARIAL DETECTION

Adversarial attacks often aim to generate samples that the model misclassifies confidently. However, these attacks may not consider the model's marginal energy, leading to overconfident adversarial samples with low negative marginal energy (See Appendix F.3 for the evidence). In models without proper uncertainty calibration, these adversarial samples can be misleading. Leveraging the insight that adversarial samples tend to reside in OOD regions, we investigated whether ECH could detect such adversarial samples effectively.

We generated adversarial samples using PGD [49], I-FGSM [39], L2 [39], and random perturbations. We then compared these adversarial samples with the ID test set to assess their distinguishability. Table 1 presents the detection scores for various types of heads. Compared to the other heads, ECH effectively distinguishes adversarial samples by treating them as uncertain OOD samples.

Table 1: Comparative results for adversarial detection depicting the mean and ($\pm$) standard deviation of performance obtained across ten individual runs.

| | PGD [49] | | I-FGSM [39] | | L2 [39] | | Random | | AVG | |
|---|---|---|---|---|---|---|---|---|---|---|
| | AUROC | AUPR | AUROC | AUPR | AUROC | AUPR | AUROC | AUPR | AUROC | AUPR |
| LinH | 79.20 | 80.92 | 79.24 | 80.93 | 84.64 | 86.42 | 85.18 | 86.89 | 82.07 | 83.79 |
| | $\pm$ 15.73 | $\pm$ 16.87 | $\pm$ 15.42 | $\pm$ 16.60 | $\pm$ 7.63 | $\pm$ 6.64 | $\pm$ 7.21 | $\pm$ 5.89 | $\pm$ 10.09 | $\pm$ 11.5 |
| CosH | 90.56 | 94.14 | 90.59 | 94.16 | 93.49 | 96.04 | 93.92 | 96.28 | 92.14 | 95.16 |
| | $\pm$ 4.40 | $\pm$ 3.13 | $\pm$ 4.37 | $\pm$ 3.06 | $\pm$ 3.57 | $\pm$ 2.06 | $\pm$ 3.40 | $\pm$ 1.95 | $\pm$ 3.78 | $\pm$ 2.55 |
| GPH | 74.35 | 69.18 | 73.73 | 68.44 | 88.98 | 89.63 | 88.96 | 88.72 | 81.51 | 78.99 |
| | $\pm$ 4.37 | $\pm$ 3.18 | $\pm$ 3.76 | $\pm$ 2.99 | $\pm$ 1.93 | $\pm$ 1.2 | $\pm$ 1.6 | $\pm$ 0.32 | $\pm$ 2.92 | $\pm$ 1.92 |
| ECH | 96.54 | 97.64 | 96.60 | 97.66 | **99.07** | **99.33** | **99.32** | **99.50** | 97.88 | **98.53** |
| ($-E(\mathbf{x})$) | $\pm$ 3.83 | $\pm$ 2.54 | $\pm$ 3.84 | $\pm$ 2.58 | $\pm$ 1.57 | $\pm$ 1.06 | $\pm$ 1.18 | $\pm$ 0.84 | $\pm$ 2.61 | $\pm$ 1.76 |
| ECH | **96.59** | **97.75** | **96.64** | **97.77** | 98.81 | 99.17 | 99.08 | 99.34 | 97.78 | 98.51 |
| ($\max_y P(y|\mathbf{x})$) | $\pm$ 3.28 | $\pm$ 2.06 | $\pm$ 3.27 | $\pm$ 2.08 | $\pm$ 1.68 | $\pm$ 1.08 | $\pm$ 1.35 | $\pm$ 0.89 | $\pm$ 2.4 | $\pm$ 1.53 |

### 6.2    OOD DETECTION

We compared our model with various baselines on OOD detection tasks with well-known six OOD benchmark datasets: SVHN [55], Texture [6], iSUN [71], LSUN [74], Places365 [79], LSUN_R [74] We train ECH with ResNet34 as the backbone model on the CIFAR-10 dataset. We use ECH's maximum conditional likelihood (MSP, [29]) and negative marginal energy (energy score, [47]) as natural OOD detection scores to validate uncertainty calibration and assess alignment between conditional uncertainty and marginal energy. Additional information regarding OOD detections is provided, including essential details regarding the selection criteria for both the baseline and benchmark datasets (Appendix F.4), statistical significance (Appendix H.6.2), and a thorough comparison of the reported results of the baselines (Appendices H.6.2 and H.6.3).

Tab. 2 compares ECH's OOD detection performance with various baselines. ECH exhibits stable performance regarding both negative marginal energy and maximum conditional likelihood. Notably, negative marginal energy demonstrates the best performance on average; there is a decrease of 7.09%

Table 2: Comparative results for OOD detection depicting the mean and (±) standard deviation of performance obtained across ten individual runs. Some baseline methods [47, 42, 64] are unable to replicate the performance; hence, the reported performances are used accordingly. Dataset: (ID) CIFAR-10; (OOD) SVHN [55], Texture [6], iSUN [71], LSUN [74], Places365 [79], LSUN_R [74]. Evaluation Metrics: False Positive Rate (FPR95) for OOD samples at a True Positive Rate (TPR) of 95%, and Area Under the Receiver Operating Characteristic Curve (AUROC).

| | SVHN[55] FPR↓ | AUROC↑ | Texture[6] FPR↓ | AUROC↑ | iSUN[71] FPR↓ | AUROC↑ | LSUN[74] FPR↓ | AUROC↑ | Places365[79] FPR↓ | AUROC↑ | LSUN_R[74] FPR↓ | AUROC↑ | AVG FPR↓ | AUC↑ | AVG w/o LSUN_R FPR↓ | AUC↑ |
|---|---|---|---|---|---|---|---|---|---|---|---|---|---|---|---|---|
| LinH | 17.67 | 96.45 | 34.47 | 90.67 | 14.58 | 97.29 | 27.91 | 93.81 | 40.12 | 88.54 | 14.86 | 97.17 | 24.94 | 93.99 | 29.27 | 92.58 |
| | ± 6.62 | ± 1.09 | ± 2.74 | ± 1.17 | ± 3.46 | ± 0.75 | ± 5.88 | ± 1.94 | ± 1.41 | ± 0.89 | ± 3.03 | ± 0.69 | ± 3.86 | ± 1.09 | ± 3.38 | ± 1.19 |
| CosH | 34.21 | 95.54 | 44.86 | 92.94 | 35.78 | 94.97 | 40.35 | 94.75 | 56.37 | 88.56 | 37.65 | 94.8 | 41.54 | 93.59 | 44.34 | 92.81 |
| | ± 12.62 | ± 1.28 | ± 2.18 | ± 0.61 | ± 5.75 | ± 0.91 | ± 6.18 | ± 0.69 | ± 1.84 | ± 0.56 | ± 5.25 | ± 0.69 | ± 5.64 | ± 0.79 | ± 3.99 | ± 0.7 |
| GPH | 99.95 | 94.3 | 99.08 | 85.56 | 99.01 | 84.66 | 99.7 | 91.35 | 98.9 | 82.37 | 98.97 | 84.71 | 99.27 | 87.16 | 99.18 | 85.99 |
| | ± 0.09 | ± 3.43 | ± 1.08 | ± 3.57 | ± 0.53 | ± 2.62 | ± 0.43 | ± 2.91 | ± 0.62 | ± 2.26 | ± 0.45 | ± 2.15 | ± 0.53 | ± 2.82 | ± 0.67 | ± 2.85 |
| MSP[29] | 31.02 | 95.236 | 46.54 | 90.311 | 30.17 | 95.53 | 43.09 | 92.503 | 52.2 | 88.027 | 30.8 | 95.406 | 38.97 | 92.84 | 43 | 91.59 |
| | ± 7.32 | ± 0.90 | ± 2.81 | ± 0.98 | ± 3.65 | ± 0.64 | ± 5.23 | ± 1.37 | ± 1.77 | ± 0.69 | ± 3.26 | ± 0.56 | ± 4.01 | ± 0.86 | ± 4.75 | ± 0.91 |
| Energy[47] | 54.41 | 91.22 | 55.23 | 89.37 | 27.52 | 95.59 | 10.19 | 98.05 | 42.77 | 91.02 | – | – | 38.02 | 93.05 | 33.93 | 93.51 |
| ODIN[44] | 44.14 | 86.75 | 53.88 | 81.09 | 13.35 | 97.27 | 14.29 | 96.92 | 50.05 | 86.13 | 11.37 | 97.67 | 31.18 | 90.97 | 32.89 | 90.35 |
| | ± 12.54 | ± 5.52 | ± 4.01 | ± 2.75 | ± 3.26 | ± 0.58 | ± 5.51 | ± 1.45 | ± 2.84 | ± 1.49 | ± 2.62 | ± 0.48 | ± 5.13 | ± 2.05 | ± 5.23 | ± 2.35 |
| GODIN[31] | 19.41 | 95.52 | 38.33 | 89.69 | 31.48 | 93.68 | 17.65 | 96.17 | 54 | 84.49 | 23.73 | 95.48 | 30.77 | 92.51 | 35.37 | 91.01 |
| | ± 5.28 | ± 1.24 | ± 5.09 | ± 3.36 | ± 9.27 | ± 2.04 | ± 9.09 | ± 3.12 | ± 5.35 | ± 3.63 | ± 5.27 | ± 1.07 | ± 6.56 | ± 2.42 | ± 6.79 | ± 2.47 |
| Mahalanobis[42] | 9.24 | 97.8 | 23.21 | 92.91 | 6.02 | 98.63 | 67.73 | 73.61 | 83.5 | 69.56 | – | – | 37.94 | 86.5 | 45.12 | 83.68 |
| ReAct[65] | 41.99 | 90.3 | 41.71 | 90.02 | 31.02 | 93.71 | 24.82 | 94.54 | 44.14 | 89.99 | 30.33 | 93.95 | 35.67 | 92.09 | 35.42 | 92.07 |
| | ± 14.87 | ± 5.01 | ± 2.56 | ± 1.53 | ± 4.66 | ± 1.09 | ± 5.76 | ± 1.82 | ± 1.61 | ± 0.48 | ± 5.95 | ± 1.60 | ± 5.91 | ± 1.93 | ± 5.29 | ± 2.18 |
| LogitNorm[70] | 17.2 | 97.16 | 33.37 | 93.32 | 19.44 | 96.72 | 2.53 | 99.38 | 33.22 | 93.32 | 17.67 | 97 | 20.57 | 96.15 | 22.14 | 95.69 |
| | ± 7.07 | ± 0.79 | ± 3.05 | ± 0.42 | ± 7.97 | ± 1.25 | ± 0.62 | ± 0.14 | ± 1.68 | ± 0.34 | ± 7.72 | ± 1.14 | ± 4.52 | ± 0.68 | ± 3.47 | ± 0.64 |
| KNN[66] | 37.66 | 94.6 | 42.02 | 93.57 | 35.61 | 94.49 | 28.63 | 89.8 | 46.52 | 91.08 | 32.03 | 95.19 | 37.08 | 93.12 | 38.2 | 92.24 |
| | ± 8.33 | ± 1.08 | ± 2.70 | ± 0.52 | ± 3.04 | ± 0.75 | ± 2.42 | ± 19.13 | ± 1.13 | ± 0.24 | ± 2.74 | ± 0.61 | ± 3.4 | ± 3.73 | ± 3.92 | ± 4.34 |
| DICE[64] | 25.29 | 95.9 | 41.9 | 88.18 | 4.36 | 99.14 | **0.02** | **99.92** | 48.59 | 89.13 | **3.91** | 99.2 | 20.68 | 95.25 | 23.72 | 94.09 |
| ECH (−E(x)) | **5.51** | **99.14** | **22.87** | **95.9** | **4.32** | **99.22** | 10.95 | 98.58 | **32.89** | **94.57** | 4.35 | **99.2** | **13.48** | **97.77** | **17.76** | **97.07** |
| | ± 1.16 | ± 0.14 | ± 2.51 | ± 0.36 | ± 1.65 | ± 0.17 | ± 4.58 | ± 0.44 | ± 4.29 | ± 1.03 | ± 1.39 | ± 0.16 | ± 2.6 | ± 0.38 | ± 2.41 | ± 0.42 |
| ECH (max_y p(y\|x)) | 7.55 | 98.92 | 25.81 | 95.73 | 6.55 | 98.87 | 14.49 | 97.65 | 37.77 | 94.2 | 6.66 | 98.81 | 16.47 | 97.36 | 21.16 | 96.61 |
| | ± 2.1 | ± 0.26 | ± 2.52 | ± 0.32 | ± 2.17 | ± 0.22 | ± 6.16 | ± 1.54 | ± 4.16 | ± 0.89 | ± 2.19 | ± 0.25 | ± 3.22 | ± 0.58 | ± 3.00 | ± 0.44 |

in FPR95 and an increase of 1.62% in AUROC compared to the best-performed baselines. This performance gain in OOD detection comes at the cost of a compromise in classification accuracy by 0.15%. ECH effectively calibrates conditional uncertainty and enhances the overall OOD detection performance. We diverged from the six established benchmarks and compared performance with seminal methods. Specifically, we evaluated the performance compared to JEM [22]. This versatile energy-based model offers a means of evaluating the marginal energy of a classifier and is additionally applicable for out-of-distribution (OOD) detection. Another method considered was SNGP [46], which shares a comparable philosophy of in-network uncertainty calibration but employs a distinct approach. The details of the performance comparison results can be found in Appendix H.6.1.

# 7 CONCLUSION

**Summary.** Unlike humans, machines often make over-confident predictions when encountering unseen samples. Theoretical analyses indicate that conditional uncertainty and marginal energy have a weak relation. By leveraging the correlation between marginal probability and conditional uncertainty observed in human decisions, we propose a novel plug-in layer, ECH, to couple marginal energy and conditional uncertainty without additional training or data. Our results demonstrate that a neural network incorporating the ECH closely replicates human-like uncertainty in OOD images, proving highly efficient in anomaly detection tasks. The most significant benefit of machine learning with human-like uncertainty is that it encourages effective human-machine cooperation and reliable AI-assisted decision-making.

**Limitation.** Despite its contributions, this study also has limitations. The experiments performed in this study are mainly limited to the computer vision task, and further investigation is required to generalize the results to other domains and data types. The effectiveness of the proposed ECH should be verified in a broader range of real-world scenarios. ECH is a preliminary step in in-network calibration, and exploring better meta-cognitive models is necessary.

**Discussion.** This research highlights the importance of understanding neural models' internal representation. Investigating the factors that affect uncertainty can provide insight into the robustness of the model. Our approach also stimulates the virtuous cycle between AI and neuroscience [26] to expand the current understanding about meta-cognition by future studies.

**Ethics Statement.** (*Societal Impact*) As our approach aims to simulate human-like uncertainty, there is a risk of increased false negatives in anomaly detection tasks. The trade-offs between over-confident and over-cautious models should be handled more carefully. If the model becomes overly cautious, it may miss identifying genuine anomalies or abnormalities in ID, leading to potential risks and

consequences. An over-cautious model can lead to deferred decisions for IDs that require accurate classification, while an over-confident model can lead to decisions about OOD with hasty confidence. Suppose, for example, that a model for distinguishing between malignant and benign tumors is trained through medical images. If the model is over-cautious, there is a risk of hesitating to make decisions for malignant tumor images that require rapid diagnosis. If the model is over-confident, images that show something other than a tumor can be confidently selected as malignant. Balancing between over-confident and over-cautious regimes is crucial for practical applicability and effectiveness. (*IRB*) All protocols for the online behavioral experiments in our study are approved by the Institutional Review Board of our institution. According to guidelines, all participants were informed about their rights (monetary rewards and arbitrary withdrawals of participation without penalty) and that the data were anonymously handled before the tutorial screen (Fig. 7a) was presented. To begin the experiment, participants must click the "Accept and Start Tutorial" button.

**Reproducibility Statement.** (*Code*) We provide the implementation code in supplementary materials to reproduce the experimental results. This includes: i) data generation for human behavioral study and verification regarding the proposed neural model, ii) behavioral experiments on the crowd-sourcing platform, iii) training of neural networks including ECH, and iv) various empirical analysis and comparative experiments. In addition to the code, we also provide instructions, dependencies, and commands. (*Datasets of Human Behavioral Study and Anonymized Human Responses*) We also provide the datasets used in human behavior experiments and the anonymized response of 250 accepted participants in supplementary materials.

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

## A  PROOFS

### A.1  PROOF OF THEOREM 1

*Proof.* **(i)** To explore the effect of $a(\mathbf{x})$ on $H(P(Y|\mathbf{x}))$, we check the derivative of $H(P(Y|\mathbf{x}))$ with respect to $a(\mathbf{x})$.

$$\frac{\partial H(P(Y|\mathbf{x}))}{\partial a(\mathbf{x})} = \sum_y \frac{\partial H(P(Y|\mathbf{x}))}{\partial P(y|\mathbf{x})} \left( \sum_{y'} \frac{\partial P(y|\mathbf{x})}{\partial f_{y'}(\mathbf{x})} \frac{\partial f_{y'}(\mathbf{x})}{\partial a(\mathbf{x})} \right)$$

Each derivative can be computed as follows:

$$\frac{\partial H(P(Y|\mathbf{x}))}{\partial P(y|\mathbf{x})} = -(1 + \log P(y|\mathbf{x})), \quad \frac{\partial P(y|\mathbf{x})}{\partial f_{y'}(\mathbf{x})} = P(y|\mathbf{x}) \cdot (\delta_{yy'} - P(y'|\mathbf{x})), \quad \frac{\partial f_y(\mathbf{x})}{\partial a(\mathbf{x})} = h_y(\mathbf{x})$$

The computations of $\frac{\partial H(P(Y|\mathbf{x}))}{\partial P(y|\mathbf{x})}$ and $\frac{\partial f_{y'}(\mathbf{x})}{\partial a(\mathbf{x})}$ are relatively trivial, while the derivative $\frac{\partial P(y|\mathbf{x})}{\partial f_{y'}(\mathbf{x})}$ can be obtained by using the following identity:

$$\frac{\partial}{\partial f_{y'}(\mathbf{x})} \log(P(y|\mathbf{x})) = \frac{1}{P(y|\mathbf{x})} \cdot \frac{\partial P(y|\mathbf{x})}{\partial f_{y'}(\mathbf{x})}$$
$$\implies \frac{\partial P(y|\mathbf{x})}{\partial f_{y'}(\mathbf{x})} = P(y|\mathbf{x}) \cdot \frac{\partial}{\partial f_{y'}(\mathbf{x})} \log(P(y|\mathbf{x}))$$

The derivative $\frac{\partial}{\partial f_{y'}(\mathbf{x})} \log(P(y|\mathbf{x}))$ can be computed as follows:

$$\frac{\partial}{\partial f_{y'}(\mathbf{x})} \log(P(y|\mathbf{x})) = \frac{\partial}{\partial f_{y'}(\mathbf{x})} \log\left(\frac{\exp(f_y(\mathbf{x}))}{\sum_{y''} \exp(f_{y''}(\mathbf{x}))}\right)$$
$$= \frac{\partial f_y(\mathbf{x})}{\partial f_{y'}(\mathbf{x})} - \frac{\partial}{\partial f_{y'}(\mathbf{x})} \log\left(\sum_{y''} \exp(f_{y''}(\mathbf{x}))\right)$$

Let $\delta_{ij} = \frac{\partial f_i}{\partial f_j}$ be the Kronecker delta. Then, we can write

$$\frac{\partial}{\partial f_{y'}(\mathbf{x})} \log(P(y|\mathbf{x})) = \delta_{yy'} - \frac{\exp(f_{y'}(\mathbf{x}))}{\sum_{y''} \exp(f_{y''}(\mathbf{x}))} = \delta_{yy'} - P(y'|\mathbf{x})$$

By plugging the derivative into $\frac{\partial P(y|\mathbf{x})}{\partial f_{y'}(\mathbf{x})}$, we obtain

$$\frac{\partial P(y|\mathbf{x})}{\partial f_{y'}(\mathbf{x})} = P(y|\mathbf{x}) \cdot \frac{\partial}{\partial f_{y'}(\mathbf{x})} \log(P(y|\mathbf{x})) = P(y|\mathbf{x}) \cdot (\delta_{yy'} - P(y'|\mathbf{x}))$$

By plugging the derivatives into $\frac{dH(P(y|\mathbf{x}))}{da(\mathbf{x})}$, we obtain

$$
\begin{aligned}
\frac{dH(P(Y|\mathbf{x}))}{da(\mathbf{x})} &= \sum_y \frac{\partial H(P(y|\mathbf{x}))}{\partial P(y|\mathbf{x})} \left( \sum_{y'} \frac{\partial P(y|\mathbf{x})}{\partial f_{y'}(\mathbf{x})} \frac{df_{y'}(\mathbf{x})}{da(\mathbf{x})} \right) \\
&= -\sum_y (1 + \log P(y|\mathbf{x})) \left( \sum_{y'} P(y|\mathbf{x}) \cdot (\delta_{yy'} - P(y'|\mathbf{x})) \cdot h_{y'}(\mathbf{x}) \right) \\
&= -\sum_y P(y|\mathbf{x}) \cdot (1 + \log P(y|\mathbf{x})) \sum_{y'} (\delta_{yy'} - P(y'|\mathbf{x})) \cdot h_{y'}(\mathbf{x}) \\
&= -\sum_y P(y|\mathbf{x}) \cdot (1 + \log P(y|\mathbf{x})) \sum_{y' \neq y} P(y'|\mathbf{x})(h_y(\mathbf{x}) - h_{y'}(\mathbf{x})) \\
&\quad \left( \text{since } \delta_{yy} = 1 = \sum_{y'} P(y'|\mathbf{x}) \right) \\
&= -\sum_y P(y|\mathbf{x}) \cdot (1 + \log P(y|\mathbf{x})) \sum_{y'} P(y'|\mathbf{x})(h_y(\mathbf{x}) - h_{y'}(\mathbf{x})) \\
&= -\left( \sum_y P(y|\mathbf{x}) \sum_{y'} P(y'|\mathbf{x})(h_y(\mathbf{x}) - h_{y'}(\mathbf{x})) \right. \\
&\quad \left. + \sum_y P(y|\mathbf{x}) \log P(y|\mathbf{x}) \sum_{y'} P(y'|\mathbf{x})(h_y(\mathbf{x}) - h_{y'}(\mathbf{x})) \right) \\
&= -\left( 0 + \sum_y P(y|\mathbf{x}) \log P(y|\mathbf{x}) \sum_{y'} P(y'|\mathbf{x})(h_y(\mathbf{x}) - h_{y'}(\mathbf{x})) \right) \\
&= -\sum_y P(y|\mathbf{x}) \log P(y|\mathbf{x})(h_y(\mathbf{x}) - \bar{h}(\mathbf{x})), \quad \text{where } \bar{h}(\mathbf{x}) = \sum_{y'} P(y'|\mathbf{x}) h_{y'}(\mathbf{x}) \\
&= -\mathbb{E}_{P(y|\mathbf{x})}[\log P(y|\mathbf{x})(h_y(\mathbf{x}) - \bar{h}(\mathbf{x}))] \\
&= -\mathbb{E}_{P(y|\mathbf{x})}[(f_y(\mathbf{x}) - \log \sum_{y''} \exp(f_{y''}(\mathbf{x})))(h_y(\mathbf{x}) - \bar{h}(\mathbf{x}))] \\
&= -\mathbb{E}_{P(y|\mathbf{x})}[f_y(\mathbf{x})(h_y(\mathbf{x}) - \bar{h}(\mathbf{x}))] \\
&\quad + \log \sum_{y''} \exp(f_{y''}(\mathbf{x})) \mathbb{E}_{P(y|\mathbf{x})}[h_y(\mathbf{x}) - \bar{h}(\mathbf{x})] \\
&= -\mathbb{E}_{P(y|\mathbf{x})}[f_y(\mathbf{x})(h_y(\mathbf{x}) - \bar{h}(\mathbf{x}))] \\
&\quad + \log \sum_{y''} \exp(f_{y''}(\mathbf{x}))(\bar{h}(\mathbf{x}) - \bar{h}(\mathbf{x})) \\
&= -\mathbb{E}_{P(y|\mathbf{x})}[f_y(\mathbf{x})(h_y(\mathbf{x}) - \bar{h}(\mathbf{x}))] + 0 \\
&= -\mathbb{E}_{P(y|\mathbf{x})}[(a(\mathbf{x})h_y(\mathbf{x}) + b_y)(h_y(\mathbf{x}) - \bar{h}(\mathbf{x}))] \\
&= -a(\mathbf{x})\mathbb{E}_{P(y|\mathbf{x})}[h_y(\mathbf{x})(h_y(\mathbf{x}) - \bar{h}(\mathbf{x}))] - \mathbb{E}_{P(y|\mathbf{x})}[b_y(h_y(\mathbf{x}) - \bar{h}(\mathbf{x}))] \\
&= -a(\mathbf{x})\left( \mathbb{E}_{P(y|\mathbf{x})}[h_y(\mathbf{x})^2] - \bar{h}(\mathbf{x})\mathbb{E}_{P(y|\mathbf{x})}[h_y(\mathbf{x})] \right) \\
&\quad - \mathbb{E}_{P(y|\mathbf{x})}[b_y(h_y(\mathbf{x}) - \bar{h}(\mathbf{x}))] \\
&= -a(\mathbf{x})\left( \mathbb{E}_{P(y|\mathbf{x})}[h_y(\mathbf{x})^2] - \mathbb{E}_{P(y|\mathbf{x})}[h_y(\mathbf{x})]^2 \right) - \mathbb{E}_{P(y|\mathbf{x})}[b_y(h_y(\mathbf{x}) - \bar{h}(\mathbf{x}))] \\
&= -a(\mathbf{x})\mathrm{Var}_{P(y|\mathbf{x})}(h_y(\mathbf{x})) - \mathbb{E}_{P(y|\mathbf{x})}[b_y\tilde{h}_y(\mathbf{x})], \quad \text{where } \tilde{h}_y(\mathbf{x}) = h_y(\mathbf{x}) - \bar{h}(\mathbf{x}) \\
&= -a(\mathbf{x})\mathrm{Var}_{P(y|\mathbf{x})}(\tilde{h}_y(\mathbf{x})) - \mathbb{E}_{P(y|\mathbf{x})}[b_y\bar{h}_y(\mathbf{x})], \quad \text{since a variance is translation invariant}
\end{aligned}
$$

In accordance with the condition $\mathbb{E}_{P(y|\mathbf{x})}[b_y\tilde{h}_y(\mathbf{x})] > -a(\mathbf{x})\mathrm{Var}_{P(y|\mathbf{x})}(\tilde{h}_y(\mathbf{x}))$, we have $\frac{dH(P(Y|\mathbf{x}))}{dP(y|\mathbf{x})} = -a(\mathbf{x})\mathrm{Var}_{P(y|\mathbf{x})}(\tilde{h}_y(\mathbf{x})) - \mathbb{E}_{P(y|\mathbf{x})}[b_y\tilde{h}_y(\mathbf{x})] < 0$. Therefore, an increase in $a(\mathbf{x})$ leads to a decrease in $H(P(Y|\mathbf{x}))$ under the condition.

Furthermore, the way in which $a(\mathbf{x})$ influences $H(P(y|\mathbf{x}))$ varies depending on the magnitude of the bias, since $H(P(Y|\mathbf{x}))$ may remain unchanged or increase if the condition does not hold.

**(ii)** Let $y^* = \arg\max_y p(y|\mathbf{x})$. The derivative of $P(y^*|\mathbf{x})$ with respect to the scaling factor $a(\mathbf{x})$ is as follows.

$$
\begin{aligned}
\frac{\partial P(y^*|\mathbf{x})}{\partial a(\mathbf{x})} &= \sum_y \frac{\partial P(y^*|\mathbf{x})}{\partial f_y(\mathbf{x})}\frac{\partial f_y(\mathbf{x})}{\partial a(\mathbf{x})} \\
&= \sum_y P(y^*|\mathbf{x})(\delta_{y^*y} - P(y|\mathbf{x}))h_y(\mathbf{x}) \\
&= P(y^*|\mathbf{x})\sum_y (\delta_{y^*y} - P(y|\mathbf{x}))h_y(\mathbf{x}) \\
&= P(y^*|\mathbf{x})\sum_{y \neq y^*} P(y|\mathbf{x})(h_{y^*}(\mathbf{x}) - h_y(\mathbf{x})) \\
&= P(y^*|\mathbf{x})\sum_y P(y|\mathbf{x})(h_{y^*}(\mathbf{x}) - h_y(\mathbf{x})) \\
&= P(y^*|\mathbf{x})(h_{y^*}(\mathbf{x}) - \bar{h}(\mathbf{x})), \text{ where } \bar{h}(\mathbf{x}) = \sum_y P(y|\mathbf{x})h_y(\mathbf{x})
\end{aligned}
$$

Since $f_{y^*}(\mathbf{x}) > \mathbb{E}_{p(y|\mathbf{x})}[f_y(\mathbf{x})]$, $a(\mathbf{x})h_{y^*}(\mathbf{x}) + b_{y^*} > a(\mathbf{x})\mathbb{E}_{p(y|\mathbf{x})}[h_y(\mathbf{x})] + \mathbb{E}_{p(y|\mathbf{x})}[b_y]$. The precedihng inequality can be re-written more concisely as $a(\mathbf{x})h_{y^*}(\mathbf{x}) + b_{y^*} > a(\mathbf{x})\bar{h}(\mathbf{x}) + \bar{b}$. Then, $h_{y^*}(\mathbf{x}) - \bar{h}(\mathbf{x}) > -\frac{b_{y^*} - \bar{b}}{a(\mathbf{x})}$ always holds. By the condition, when $b_{y^*} \leq \bar{b}$ and $a(\mathbf{x}) > 0$, $h_{y^*}(\mathbf{x}) - \bar{h}(\mathbf{x}) > -\frac{b_{y^*} - \bar{b}}{a(\mathbf{x})} \geq 0$, and thus $P(y^*|\mathbf{x})(h_{y^*}(\mathbf{x}) - \bar{h}(\mathbf{x})) > 0$. Therefore, $P(y^*|\mathbf{x})$ increases with respect to $a(\mathbf{x})$ when $b_{y^*} \leq \bar{b}$.

**(iii)** The derivative of $E(\mathbf{x})$ with respect to the scaling factor $a(\mathbf{x})$ is as follows.

$$
\frac{dE(\mathbf{x})}{da(\mathbf{x})} = \sum_y \frac{\partial E(\mathbf{x})}{\partial f_y(\mathbf{x})}\frac{df_y(\mathbf{x})}{da(\mathbf{x})}
$$

The marginal energy $E(\mathbf{x})$ is defined as $-\log\sum_y \exp(f_y(\mathbf{x}))$. Then, the derivative of marginal energy with respect to logit becomes $\frac{\partial E(\mathbf{x})}{\partial f_y(\mathbf{x})} = -\frac{\exp(f_y(\mathbf{x}))}{\sum_{y'}\exp(f_{y'}(\mathbf{x}))} = -P(y|\mathbf{x})$. In addition, the derivative of logit with respect to scaling factor is $\frac{\partial f_y(\mathbf{x})}{\partial a(\mathbf{x})} = h_y(\mathbf{x})$. Then, we have

$$
\begin{aligned}
\frac{\partial E(\mathbf{x})}{\partial a(\mathbf{x})} &= \sum_y \frac{\partial E(\mathbf{x})}{\partial f_y(\mathbf{x})}\frac{\partial f_y(\mathbf{x})}{\partial a(\mathbf{x})} \\
&= -\sum_y P(y|\mathbf{x})h_y(\mathbf{x}) \\
&= -\mathbb{E}_{P(y|\mathbf{x})}[h_y(\mathbf{x})] := -\bar{h}(\mathbf{x}).
\end{aligned}
$$

Therefore, if $\bar{h}(\mathbf{x}) > 0$, as $a(\mathbf{x})$increase, $E(\mathbf{x})$ decreases. $\qquad\square$

### A.2  PROOF OF THEOREM 2

Lemma 1 shows that two samples with different marginal energies can exhibit equivalent conditional probability and, thus, uncertainty, including entropy $H(P(Y|\mathbf{x}))$ and maximum conditional likelihood $\max_y P(y|\mathbf{x})$.

**Lemma 1.** *Let $\mathbf{x}$ and $\mathbf{x}'$ be two samples. Suppose that $f_y(\mathbf{x}) = f_y(\mathbf{x}') + c$ for some $c \in \mathbb{R}$ and all $y \in \mathcal{Y}$. Then, $-E(\mathbf{x}) = c - E(\mathbf{x}')$ and $p(y|\mathbf{x}) = p(y|\mathbf{x}')$ for all $y$.*

*Proof.* Considering that the two samples $\mathbf{x}$ and $\mathbf{x}'$ are related as $f_y(\mathbf{x}) = f_y(\mathbf{x}') + c$, the conditional likelihood of them has the following relation:

$$
\begin{aligned}
P(y|\mathbf{x}) &= \frac{\exp(f_y(\mathbf{x}))}{\sum_{y'} \exp(f_{y'}(\mathbf{x}))} \\
&= \frac{\exp(f_y(\mathbf{x}') + c)}{\sum_{y'} \exp(f_{y'}(\mathbf{x}') + c)} \\
&= \frac{\exp(f_y(\mathbf{x}')) \exp(c)}{\sum_{y'} \exp(f_{y'}(\mathbf{x}')) \exp(c)} \\
&= \frac{\exp(c) \exp(f_y(\mathbf{x}'))}{\exp(c) \sum_{y'} \exp(f_{y'}(\mathbf{x}'))} \\
&= \frac{\exp(f_y(\mathbf{x}'))}{\sum_{y'} \exp(f_{y'}(\mathbf{x}'))} = P(y|\mathbf{x}')
\end{aligned}
$$

This implies that the conditional likelihood is invariant under the joint energy translation.

Meanwhile, the negative marginal energy of $\mathbf{x}$ and $\mathbf{x}'$ has following relation:

$$
\begin{aligned}
-E(\mathbf{x}) &= \log \sum_y \exp(f_y(\mathbf{x})) \\
&= \log \sum_y \exp(f_y(\mathbf{x}') + c) \\
&= \log \sum_y \exp(f_y(\mathbf{x}')) \exp(c) \\
&= \log \exp(c) \sum_y \exp(f_y(\mathbf{x}')) \\
&= \log \exp(c) - \log \sum_y \exp(f_y(\mathbf{x}')) \\
&= c - E(\mathbf{x}')
\end{aligned}
$$

Therefore, the conditional likelihood is invariant under the joint energy translation, but the translation is fully reflected in the marginal energy. □

The attributes of Lemma 1 stem from the softmax function converting the model's energy into probabilities. Nonetheless, instances where sample pairs adhere to an energy translation relation can be considered exceptionally rare. As an alternative, Lemma 2 show that samples associated with different amounts of marginal energies can have the same conditional uncertainty value, albeit under more relaxed conditions.

**Lemma 2.** *Let us assume the presence of two samples $\mathbf{x}$ and $\mathbf{x}'$, where $E(\mathbf{x}) \neq E(\mathbf{x}')$. Under the following conditions, the conditional uncertainties of the samples are the same:*

   *(i) If $E(\mathbf{x}) - E(\mathbf{x}') = \mathbb{E}_{P(y|\mathbf{x})}[E(\mathbf{x}, y)] - \mathbb{E}_{P(y|\mathbf{x}')}[E(\mathbf{x}', y)]$, then $H(P(Y|\mathbf{x})) = H(P(Y|\mathbf{x}'))$.*

   *(ii) If $E(\mathbf{x}) - E(\mathbf{x}') = E(\mathbf{x}, y^*) - E(\mathbf{x}', y^\star)$, where $y^* = \arg\max_y p(y|\mathbf{x})$ and $y^\star = \arg\max_y p(y|\mathbf{x}')$, then $p(y^*|\mathbf{x}) = p(y^\star|\mathbf{x}')$.*

*Proof.* **(i)** Let $y^* = \arg\max_y P(y|\mathbf{x})$ and $y^\star = \arg\max_y P(y|\mathbf{x}')$. We can expand the conditions regarding the deviation of marginal energy as follows.

$$E(\mathbf{x}) - E(\mathbf{x}') = E(\mathbf{x}, y^*) - E(\mathbf{x}', y^\star)$$

$$\implies -\log\sum_y \exp(f_y(\mathbf{x})) + \log\sum_{y'} \exp(f_{y'}(\mathbf{x}')) = -f_{y^*}(\mathbf{x}) + f_{y^\star}(\mathbf{x}')$$

$$\implies f_{y^*}(\mathbf{x}) - \log\sum_y \exp(f_y(\mathbf{x})) = f_{y^\star}(\mathbf{x}') - \log\sum_{y'}\exp(f_{y'}(\mathbf{x}'))$$

$$\implies \exp(f_{y^*}(\mathbf{x}) - \log\sum_y \exp(f_y(\mathbf{x}))) = \exp(f_{y^\star}(\mathbf{x}') - \log\sum_{y'}\exp(f_{y'}(\mathbf{x}')))$$

$$\implies \frac{\exp(f_{y^*}(\mathbf{x}))}{\sum_y \exp(f_y(\mathbf{x}))} = \frac{\exp(f_{y^\star}(\mathbf{x}'))}{\log\sum_{y'}\exp(f_{y'}(\mathbf{x}'))}$$

$$\implies P(y^*|\mathbf{x}) = P(y^\star|\mathbf{x}')$$

Hence, despite a deviation in marginal energy, the maximum of conditional likelihood may remain unchanged.

**(ii)** We can expand another condition regarding the deviation of marginal energy as follows.

$$E(\mathbf{x}) - E(\mathbf{x}') = \mathbb{E}_{P(y|\mathbf{x})}[E(\mathbf{x},y)] - \mathbb{E}_{P(y'|\mathbf{x}')}[E(\mathbf{x}',y')]$$

$$\implies -\log\sum_{y''}\exp(f_{y''}(\mathbf{x})) + \log\sum_{y'''}\exp(f_{y'''}(\mathbf{x}'))$$

$$= \sum_y P(y|\mathbf{x})(-f_y(\mathbf{x})) - \sum_{y'} P(y'|\mathbf{x}')(-f_{y'}(\mathbf{x}'))$$

$$\implies \sum_y P(y|\mathbf{x})f_y(\mathbf{x}) - \log\sum_{y''}\exp(f_{y''}(\mathbf{x}))$$

$$= \sum_{y'} P(y'|\mathbf{x}')f_{y'}(\mathbf{x}') - \log\sum_{y'''}\exp(f_{y'''}(\mathbf{x}'))$$

$$\implies \sum_y P(y|\mathbf{x})(f_y(\mathbf{x}) - \log\sum_{y''}\exp(f_{y''}(\mathbf{x})))$$

$$= \sum_{y'} P(y'|\mathbf{x}')(f_{y'}(\mathbf{x}') - \log\sum_{y'''}\exp(f_{y'''}(\mathbf{x}')))$$

$$\implies \sum_y P(y|\mathbf{x})(\log\exp(f_y(\mathbf{x})) - \log\sum_{y''}\exp(f_{y''}(\mathbf{x})))$$

$$= \sum_{y'} P(y'|\mathbf{x}')(\log\exp(f_{y'}(\mathbf{x}')) - \log\sum_{y'''}\exp(f_{y'''}(\mathbf{x}')))$$

$$\implies \sum_y P(y|\mathbf{x})\log\frac{\exp(f_y(\mathbf{x}))}{\sum_{y''}\exp(f_{y''}(\mathbf{x}))} = \sum_{y'} P(y'|\mathbf{x}')\log\frac{\exp(f_{y'}(\mathbf{x}'))}{\sum_{y'''}\exp(f_{y'''}(\mathbf{x}'))}$$

$$\implies \sum_y P(y|\mathbf{x})\log P(y|\mathbf{x}) = \sum_{y'} P(y'|\mathbf{x}')\log P(y'|\mathbf{x}')$$

$$\implies H(P(Y|\mathbf{x})) = H(P(Y|\mathbf{x}'))$$

Thus, the entropy can remain the same even with a deviation in marginal energy. □

Thm. 2 is followed by Lemmas 1 and 2.

## B   EMPIRICAL VALIDATION OF THEOREMS

We examine whether the conditions stipulated in Thms. 1 and 2 are satisfied in Fig. 6. Although we set a certain direction of $a(\mathbf{x})$ influence in Thm. 1, $a(\mathbf{x})$ affects the measures of uncertainty and

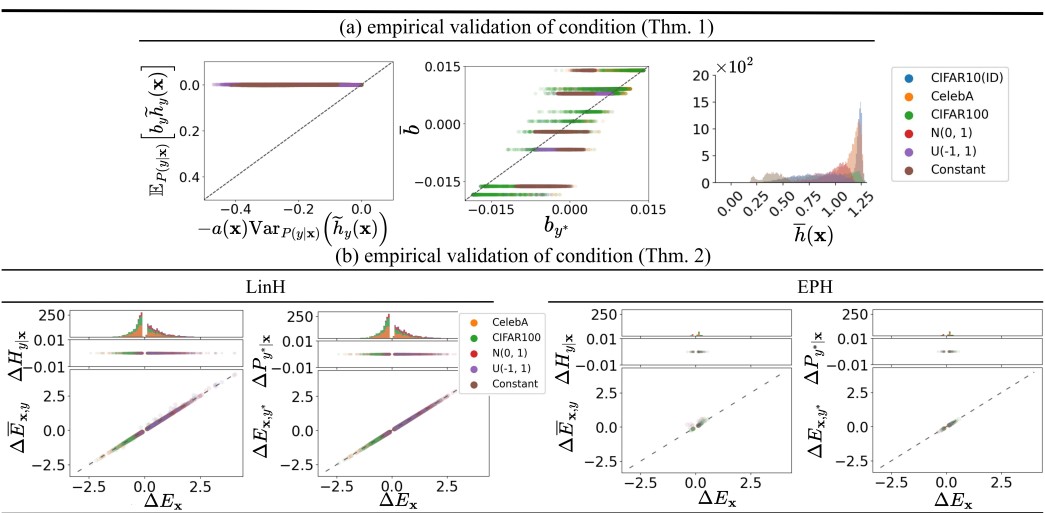

Figure 6: Empirical validation of conditions of Thms. 1 and 2 conducted using the models with LinH and ECH trained on ID (CIFAR-10) samples. Datasets that have not been used to assess OOD detection are selected; ID: CIFAR-10 [38], OOD: CIFAR-100, CelebA [48], Gaussian Noise $\mathcal{N}(0,1)$, Uniform Noise $U(-1,1)$, Constant (See Appendix F.4 for the details). (a:left-right) shows the conditions of Thm. 1 (i)-(iii). (b) presents the observation of sample pairs satisfying Thm. 2; each plot is a stack of three subplots, where the horizontal axis represents $\Delta E_{\mathbf{x}} = E(\mathbf{x}_{ID}) - E(\mathbf{x}_{OOD})$; (top) exhibits a histogram, (middle) displays (left) $\Delta H_{y|\mathbf{x}} = H(P(y|\mathbf{x}_{ID})) - H(P(y|\mathbf{x}_{OOD}))$ or (right) $\Delta P_{y^*|\mathbf{x}} = P(y^*|\mathbf{x}_{ID}) - P(y^\star|\mathbf{x}_{OOD})$, influencing the sample pairs selection, and (bottom) showcases $\Delta \bar{E}_{\mathbf{x},y} = \mathbb{E}_{P(y|\mathbf{x}_{ID})}[E(\mathbf{x}_{ID},y)] - \mathbb{E}_{P(y|\mathbf{x}_{OOD})}[E(\mathbf{x}_{OOD},y)]$ and $\Delta E_{\mathbf{x},y^*} = E(\mathbf{x}_{ID}, y^*) - E(\mathbf{x}_{OOD}, y^\star)$ for the sample pairs; (left two) for LinH, (right two) for ECH, validating (i) and (ii) of Thm. 2.

marginal energy positively or negatively unless both sides of a condition in each statement in Thm. 1 are the same. The empirical evidence presented in Fig. 6(a) further substantiates the effect of the latent norm on uncertainty and marginal energy. Although the effect does not necessarily have to be adverse, it can limit the uncertainty calibration process. The observations in Fig. 6(b) confirm the view; there exists a considerable number of sample pairs satisfying the conditions with the model with LinH in Thm. 2, while the ECH effectively eliminates such cases.

## C  THEORETICAL ANALYSIS FOR DISTANT SAMPLES

Notably, the latent norm tends to increase in faraway samples in the ReLU [17, 18, 54] network in Thm. 2. This increase in latent norm can induce overconfidence by acting as a scaling factor for energy.

In the research conducted by [28], an intriguing observation arises regarding the behavior of the conditional likelihood depending on the input in a ReLU network. Specifically, when an input moves away from the origin in a predefined direction within a ReLU network, it eventually resides within an infinitely extending polytope. This characteristic has been linked to the network's overconfidence. To prove the theorems that we claim, we will refer to a proposition presented in the study and call it Lemma 3.

**Lemma 3.** *[28] Let $\{Q_r\}_{r=1}^R$ be a set of convex polytopes on which a ReLU-network $g : \mathbb{R}^{d_0} \to \mathbb{R}^{d_l}$ is an affine function, that is for every $k \in \{1, \cdots, R\}$ and $\mathbf{x} \in Q_k$ there exists $V^k \in \mathbb{R}^{d_l \times d_0}$ and $\mathbf{c}^k \in \mathbb{R}^{d_l}$ such that $g(\mathbf{x}) = V^k \mathbf{x} + \mathbf{c}^k$. For any $\mathbf{x} \in \mathbb{R}^d$ with $\mathbf{x} \neq \mathbf{0}$, there exists $\alpha > 0$ and $t \in \{1, \cdots, R\}$ such that $\beta \mathbf{x} \in Q_t$ for all $\beta \geq \alpha$.*

**Theorem 3.** *Let $\mathbf{x} \in \mathbb{R}^d$ with $\mathbf{x} \neq \mathbf{0}$ and $g : \mathbb{R}^{d_0} \to \mathbb{R}^{d_l}$ be a ReLU-encoder mapping from input space to latent space. Let $\mathbf{z}_\beta = g(\beta \mathbf{x})$. If $\nabla_\beta g(\beta \mathbf{x}) \neq 0$, then $\lim_{\beta \to \infty} \|\mathbf{z}_\beta\| = \infty$.*

*Proof.* By Lemma 3, there exists a set of convex polytopes $\{Q_r\}_{r=1}^R$ on each of which $g$ is an affine function, that is for every $k \in \{1, \cdots, R\}$ and $\mathbf{x} \in Q_k$ there exist $U \in \mathbb{R}^{d_l \times d_0}$ and $\mathbf{d} \in \mathbb{R}^{d_l}$ such that $g(\mathbf{x}) = U\mathbf{x} + \mathbf{d}$. Denote by $Q_t$ the polytope such that $\beta \mathbf{x} \in Q_t$ for all $\beta \geq \alpha$ and let $\mathbf{z}_\beta = U(\beta \mathbf{x}) + \mathbf{d}$ with $U \in \mathbb{R}^{d_l \times d_0}$ and $\mathbf{d} \in \mathbb{R}^{d_l}$ be the latent vector with encoder $g$. Since $\nabla_\beta g(\beta \mathbf{x}) = \nabla_\beta (U(\beta \mathbf{x}) + \mathbf{d}) = U\mathbf{x}$, $\nabla_\beta g(\beta \mathbf{x}) \neq 0$ implies $U\mathbf{x} \neq 0$. Obseve that $\|\mathbf{z}_\beta\| = \|U(\beta \mathbf{x}) + \mathbf{d}\| = \|\beta U\mathbf{x} + \mathbf{d}\| \geq \|\beta U\mathbf{x}\| - \|\mathbf{d}\|$ by triangular inequality. Since $\lim_{\beta \to \infty} \|\beta U\mathbf{x}\| = \lim_{\beta \to \infty} \beta \|U\mathbf{x}\| = \infty$ for $U\mathbf{x} \neq 0$, $\lim_{\beta \to \infty} \|\mathbf{z}_\beta\| = \lim_{\beta \to \infty} \|\beta U\mathbf{x} + \mathbf{d}\| = \infty$. $\square$

## D  DETAILS OF HUMAN BEHAVIORAL EXPERIMENT

Measuring and quantifying human decision confidence (subject's internal belief about the accuracy of the decision) and metacognition have a long history in cognitive psychology and neuroscience [59]. Recently there have been many attempts to compare human and machine classifiers' decisions for ID [2] and OOD samples [20, 80], but little is known about the human confidence responses to those manipulations. Although conventional paradigms collecting confidence from humans or animals include direct self-report (rating), across-trial decision variability, and wagering (betting) [50], it remains technically tricky to quantify such a subjective measure due to substantial individual differences. To detour this problem we crowd-sourced the human classification decisions for the ID and OOD samples, and used the distributions of decisions as a population-level proxy for the confidence for the given sample (see Sec. B.3 below for detail). This section provides detailed information for our experiments and analyses employed Python packages, such as PyTorch [57], NumPy [25], and SciPy [68], for the reproducibility of our work.

### D.1  DATASET

We used the CIFAR-10 [38] dataset, which is most widely used in OOD detection studies. The dataset comprises ten classes, and 50,000 training and 10,000 test images, respectively, and we only used the test set images. We created ten exclusive sets of images, each of which has 125 images, where twenty-five of them are ID samples used for attention checks. The remaining 100 images in each set were used to generate the samples, resulting in a total of 1,000 images used throughout the experiment. Images were chosen as balanced as possible in terms of class labels and

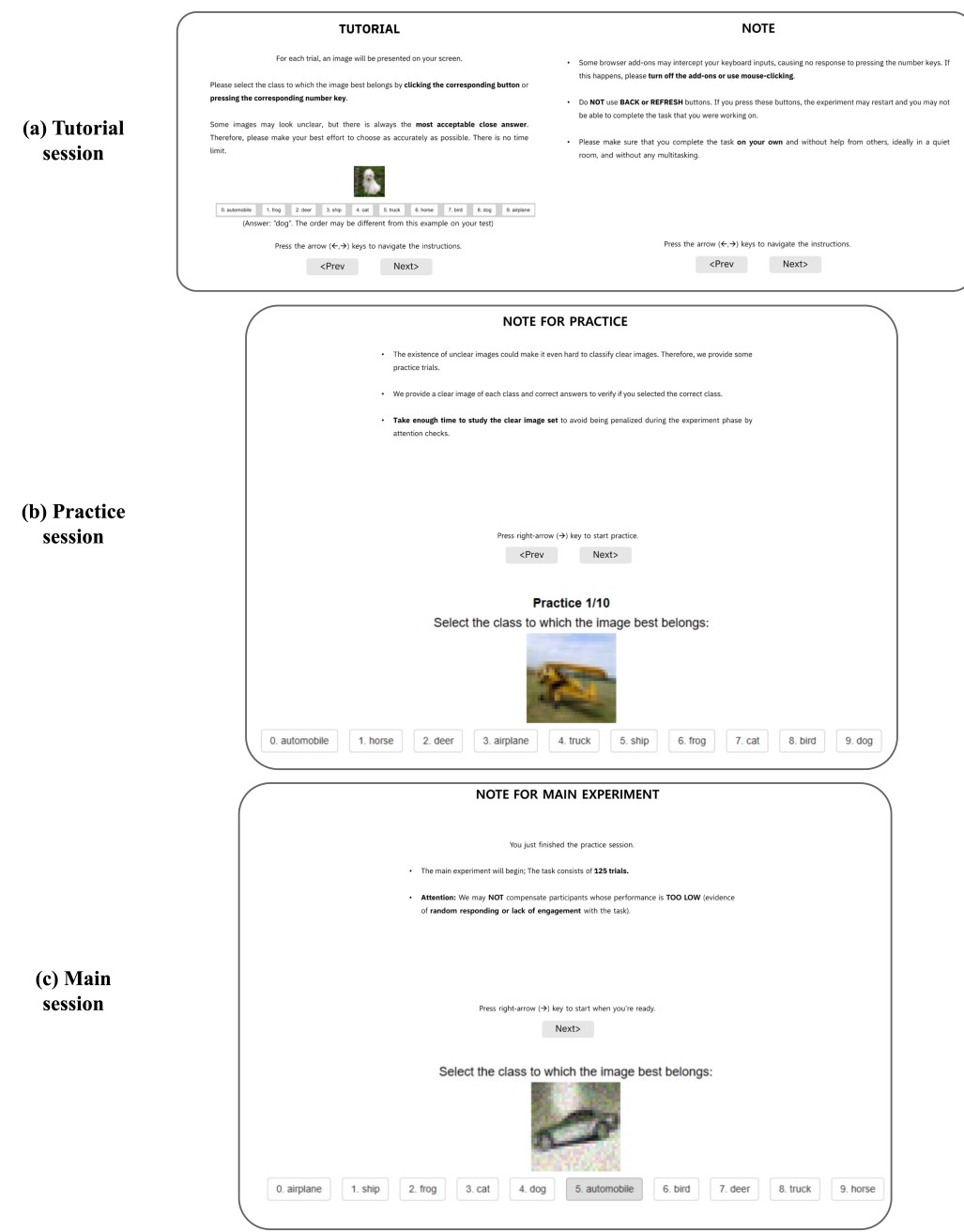

Figure 7: Schematic plot of behavioral study

lambda values. An identical set of 1,000 images was used both in the behavioral experiment analyses and the neural network simulations. To generate OOD images, random noises were added on the corresponding ID samples using PyTorch, as described in the main text: $\mathbf{x}' = (1 - \lambda)\mathbf{x}_i + (\lambda)\mathbf{n}$, where $\mathbf{x}_i \in \mathcal{D}, \mathbf{n} \sim U(0, 1)^{W \times H \times D}$, and $\lambda \in [0, 0.9]$. Additionally, images, which were initially $32 \times 32$ pixels, were resized to $128 \times 128$ pixels. The selection of $\lambda$ values was influenced by an approach analogous to that employed in [21, 20]. These values are log-scaled to compress the range of low $\lambda$ values that change more sensitively to changes in this parameter.

## D.2 BEHAVIORAL STUDY

**Online Experimental Platform**    The online experiment was delivered by Cloud Research [45] [1] and Connect [2] service to recruit and manage the participants and upload web-based experiments. The web experiment was developed using Flask [24], a Python-based web framework, and deployed using Gunicorn. All experiments underwent in a full-screen mode, and the behavioral data were saved as a JSON file through the Cloud Research platform.

**Subjects**    Two hundred fifty subjects were recruited for the experiment. One image set was assigned to twenty-five subjects. A particular ID or OOD sample was repeatedly tested by twenty-five people, whereas the subjects did not experience the same image twice. Every subject gave informed written consent forms and all procedures were approved by the Institutional Review Board.

**Compensations**    Under our study design, we uniquely assigned each task to an individual participant, preventing re-entry for those who had previously participated. Accordingly, the compensation per task is 1.5 USD, with equal compensation attributed per participant (Sec 2.2 and line 124).

**Procedure**    The web experiment encompassed three sessions: tutorial, practice, and main sessions, as shown in Fig. 7. During the tutorial session, the participants were provided with a short instruction about the task. To help the task comprehension, participants practiced the task with ten sample images without noise taken from each of the ten CIFAR-10 categories, which did not appear in the main session. During the practice session, participants were allowed to retry in case of incorrect classifications, and the main session did not begin until they passed all practice trials. The main session consisted of 125 trials where subjects were required to choose the names of categories by clicking or inputting the number keys to proceed to the next trial. Image classes appeared randomly. To check the participants' attention level during the task we presented 25 trials at random intervals with ID samples. All passed the attention check criterion, 80% of accuracy in these trials which is determined prior to the experiment (the lowest accuracy for the attention check trials was 83%). The total duration of the experiment, including all sessions, had a 30-minutes time limit, but most of the experiments took approximately 10 to 15 minutes.

## D.3 ANALYSIS DETAILS AND ADDITIONAL RESULTS

**Human Classifiers**    Our main interest is to monitor how the distributions of human decisions, not the accuracy itself, change over the OOD noise levels (Fig. 8a). To do this, we plotted histograms of the subjects' decisions for the given test image categories after normalizing them by the total number of test images, which is a conditional probability distribution over categories. Expectedly, when no noise, *i.e.,* for ID samples, the probabilities of choosing a certain class extremely diverge, near either zero (for wrong class) or one (for true class), and gradually converge to the level close to the chance $P_{\text{chance}} = 0.1$ as the noise increase (Fig. 8b, blue bars from the left to the right panels), irrespective of the true labels (Fig. 8c, top panels). Such convergence pattern can also be tracked by the max probability over classes, $P_{\text{Max}}$, which decreases as $\lambda$ increases (Fig. 8b, blue dashed lines). For a more comprehensive measure, instead of taking a single probability value, we also used the entropy of the probability distributions and confirmed similar findings (see Sec. 2.2 and Sec. 5.1 in the main text).

**Neural Network Classifiers**    Our study employed the WRN-28-10 network trained on the CIFAR-10. The baseline model, LinH, and the proposed model, ECH, were utilized in our experiments. For more detailed information regarding the implementation and training of the networks, please refer to Appendix C. The decision probability of the network was computed using its conditional likelihood for the same image set, which is used in human behavior experiments.

Like in human experiments, we analyzed the probability distributions at varying OOD noise levels. ECH showed widely distributed probabilities, resulting in smaller $P_{\text{Max}}$ compared to that of LinH (Fig. 8b,c).

---

[1]https://www.cloudresearch.com/
[2]https://connect.cloudresearch.com/

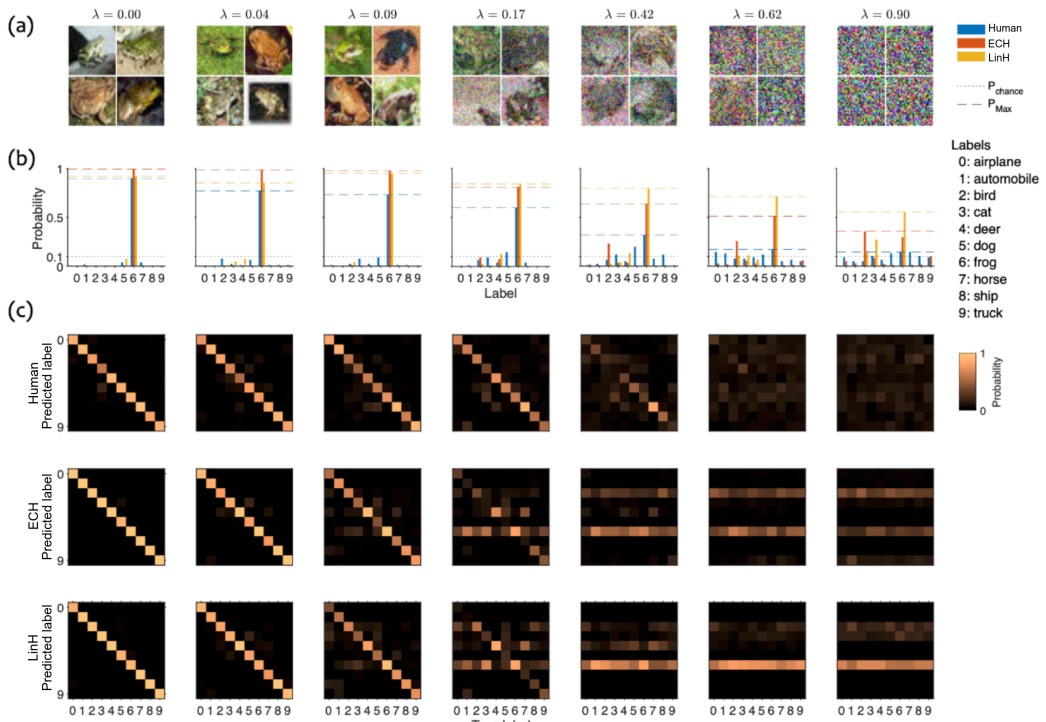

Figure 8: Additional detailed results. (a) Four randomly selected example images from a 'Frog' category (label number 6 in (b)). (b) The probability of the human and neural network classifiers averaged across all ID or OOD samples in the 'Frog' category. The dotted line is the chance level, $P_{\text{chance}}$. Dashed lines denote $P_{\text{Max}} = max_{y'}(p(y'|\mathbf{x}))$. (c) The probabilities for all categories and classifiers. (b) is from the vertical cross-sections at the location of the true label 6.

### D.4  HUMAN fMRI EXPERIMENTS

**Human fMRI experiments**   To test whether logit and latent layers exhibit similar statistical properties with human cortical activities, we conducted separate fMRI experiments and performed representational similarity analysis (Kriegeskorte et al. (2008); Fig. 9). For the model, we computed logit and latent layers activities for a given input image, then computed Spearman's rank correlation between input stimulus category pairs. Similar procedures were done in fMRI activities from brain areas, except that activities in fMRI data are obtained from the t-values from standard univariate GLM analysis. The remaining procedures follow the work done by Kriegeskorte et al. (2008). We found that logit and latent layers in ECH show higher variability in RSM values at large perturbation conditions (large $\lambda$) compared to the penultimate latent layers(Fig. 9). However, such a difference between layers was not found in LinH. In a similar analogy, IT shows higher standard deviations of RSM values than early sensory area V1 for large $\lambda$ conditions. Such a finding confirms that ECH has similar statistical properties with the cortex.

### E  DETAILS OF NEURAL MODEL IMPLEMENTATION

**Code**   We provide the implementation code to reproduce the experimental results. This includes: i) data generation for human behavioral study and verification regarding the proposed neural model, ii) behavioral experiments on the crowd-sourcing platform, iii) training of neural networks including ECH, and iv) various empirical analysis and comparative experiments. In addition to the code, we also provide instructions, dependencies, and commands.

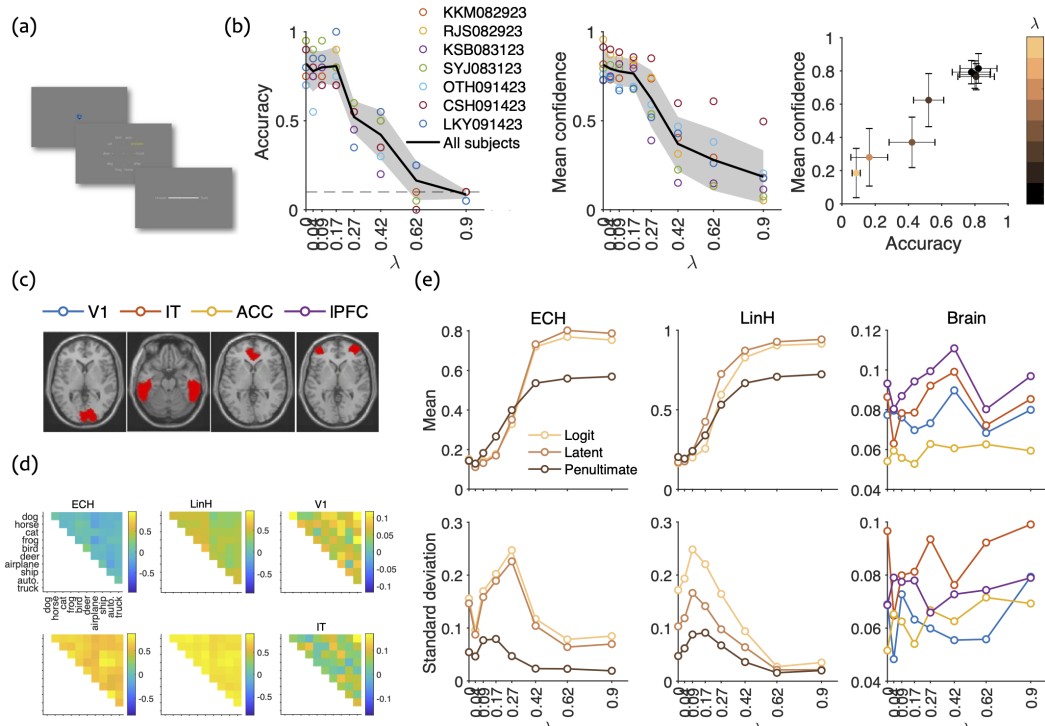

Figure 9: fMRI experiment procedure and result. (a) After seeing a subset of the ID and OOD samples (160 images per subject total, identical image sets across subjects) used in the online behavioral experiments, seven participants classified images one out of the categories and reported their confidence. (b) Accuracy and reported confidence results were reproduced as found in Fig 1 a,b. (c) We measured fMRI activities from visual task-related brain areas processing stimulus categories and decision-making. (d) Example representational similarity matrices (RSMs) from the model (left two columns; upper ones for low $\lambda$ conditions and the lower ones for high $\lambda$ conditions respectively) and brain areas (right columns; averaged across all $\lambda$ conditions). (e) Statistics of RSM off-diagonal elements. Upper panels, the mean of the absolute values of RSM in each $\lambda$ condition. Lower panels, the standard deviation of the RSM values in each $\lambda$ condition. Right most panels, the color legend is shown in panel (a). All data points represent across subject average.

**Model Architecture**    We employed the WRN-28-10 model as the basis for our analysis. In the context of OOD detection, we conducted experiments by three different architectures: WRN-28-10, ResNet18, and ResNet34 (See Appendix G.2). We used publicly available code for the models.

**Training**    During the training process, we employed popularly adopted hyperparameters to train the neural networks and did not conduct separate tuning procedures. Specifically, we trained the models for 200 epochs. The learning rate got warm-up [27] for the first 1000 iterations, then starting at 0.1 and decreasing by 0.2 at the 60th, 120th, and 160th epochs. We utilized a batch size of 128, and we employed the SGD optimizer. The weight decay for SGD was set to $5 \times 10^{-4}$, and the momentum was set to 0.9.

**Implementation of ECH**    ECH is a novel neural module comprising three components. We selected the hyperparameters of ECH that yielded the most stable results (See Appendix G.2). More specifically, the projection map $\phi$ in the kernel is an MLP involving a single hidden layer, and the non-linear activation function incorporates SeLU [37] and projects the latent $\mathbf{z}$ into a 64-dimensional space.

**Other Heads**

- LinH utilizes the linear layer provided by PyTorch.

- CosH involves a dot product between the normalized weight and a latent variable.
- GPH uses a GP layer provided by GPyTorch [19]

# F  EXPERIMENTAL DETAILS

We provide comprehensive information regarding the experiments in the main paper.

## F.1  MODEL CALIBRATION

We compute the conditional likelihood for each model for the CIFAR-10 test set. We then divided the interval of conditional likelihood into 0.05 units. The population-level human decision probability and average accuracy of samples for each interval are calculated to generate the calibration plots. The ECE is determined by the sum of the difference (or error) between conditional likelihood and human decision probability (or accuracy) weighted by the frequency ratio of samples.

## F.2  UNCERTAINTY CALIBRATION FOR DISTANT SAMPLES

We performed an analysis to validate the efficacy of ECH in calibrating the conditional uncertainty. A set of 1,000 randomly selected CIFAR-10 images was utilized. The direction vectors are obtained for a given image by normalizing two random vectors sampled from the Spherical Gaussian distribution. That is, $\mathbf{x}' = \mathbf{x}_i + c_1\bar{\mathbf{u}}_1 + c_2\bar{\mathbf{u}}_2$, where $\mathbf{x}_i \in \mathcal{D}$, $\bar{\mathbf{u}}_j = \frac{\mathbf{u}_j}{\|\mathbf{u}_j\|}$, $\mathbf{u}_j \sim N(0, I)$, $I \in \mathbb{R}^{W \times H \times D}$, $c_j$ is a coefficient of grid, and $j \in \{1, 2\}$. Subsequently, a 40 by 40 grid centered on the image was generated, scaled at $[-10^0, 10^0], [-10^1, 10^1], [-10^2, 10^2]$. The set of samples $\mathbf{x}'$ is fed into the model to obtain the conditional likelihood. This process is iterated for 1,000 input grids, and the entropy over 1,000 trials for each grid point is averaged for reporting.

## F.3  ADVERSARIAL DETECTION

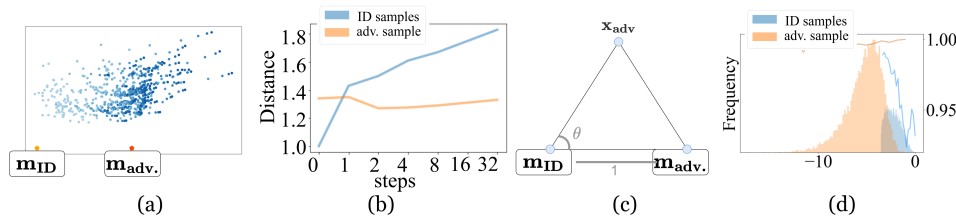

Figure 10: Adversarial samples (a) move away from ID during the perturbation process, (b) presents the average distance from class conditional means $\mathbf{m}_{ID}$ and $\mathbf{m}_{adv}$, and (c) shows a schematic plot of how we draw adversarial samples relative to class conditional means $\mathbf{m}_{ID}$ and $\mathbf{m}_{adv}$ in two-dimensional spaces. (d) shows that the samples lie in low-density regions, where the density is estimated by kernel density estimation in latent space.

This section aims to show the relationship between adversarial vulnerability and overconfidence. We show that adversarial samples are located in the OOD region when we use the LinH model (WRN-28-10). We performed a PGD [49] attack (eps=8/255) on the test set trained on CIFAR10 data. We examined whether adversarial samples moved away from the original class's latent cluster to the target class's latent cluster during the adversarial perturbation process. Fig. 10 (a) shows trajectories of the adversarial samples. Instead of heading directly toward the conditional cluster (mean = $\mathbf{m}_{adv}$) of the attack class, they pass it and continue moving further away. Fig. 10 (b) shows the average distance from class conditional mean $\mathbf{m}_{adv}$ decreases for a while and then increases again, while the average distance from class conditional mean $\mathbf{m}_{ID}$ increases. In addition, a systematic plot is displayed to monitor the trajectory of adversarial perturbation in the latent space shown in Fig. 10 (c). The class conditional latent mean is computed using the ID samples. Subsequently, the distance between each class conditional mean for the adversarial class (i.e., the class predicted by the model for the adversarial sample) and the original class is calculated, denoted by $\mathbf{m}_{ID}$ and $\mathbf{m}_{adv}$, respectively. We represent the position of the adversarial sample in a two-dimensional space

determined by three points, $\mathbf{m}_{ID}$, $\mathbf{m}_{adv}$, and adversarial sample, utilizing the normalized length of the triangle with three vertices normalized by the distance between $\mathbf{m}_{ID}$ and $\mathbf{m}_{adv}$. This process is performed iteratively, examining the trajectory at each step. Fig. 10 (d) shows a histogram based on the (Gaussian) kernel density of the latent vectors of ID samples and a line plot of the average conditional likelihood of each bin. We observed that the LinH exhibits overconfidence for adversarial samples (line plot) but is mainly distributed in a region with low kernel density in latent space. This characteristic resembles that of OOD samples. We estimated the log densities of adversarial and ID samples using the Gaussian kernel density of ID samples in the latent space. We translate log density so that the maximum value becomes zero and then represent it as a histogram. Additionally, the means of the conditional likelihood of both adversarial and ID samples for each bin in the histogram are represented as line plots.

### F.4 OOD DETECTION

Each head is independently trained ten times on the ID dataset for performance comparison experiments. The results in the table within the text represent the mean and standard deviation obtained from these experiments.

**Selection Criteria of Baselines**   The baseline selection criteria were intentionally designed to exclude methodologies reliant on synthetic data augmentation or distinct learning mechanisms. Furthermore, the selected baselines specifically address performance across six representative OOD benchmark datasets. We extracted baseline method results for out-of-distribution (OOD) detection from each paper. Certain studies omitted results for LSUN-R, leading to their exclusion from consideration. We plan to present outcomes excluding LSUN-R to address potential bias. The statistical significance of our investigation was established through a comparative analysis in Appendix H.6.2.

**OOD Datasets in Fig. 2**   In our analysis, the CIFAR-10 dataset served as the ID dataset, while the CIFAR-100 dataset, CIFAR-10 interpolation, N, U, OODom, and Constant datasets were utilized as OOD datasets. To ensure compatibility, the input images are converted to a range of $[-1, 1]$, and revised standardization is performed using statistics of the ID dataset before the first layer in the network. The dataset N consists of 10,000 images, with each pixel value sampled from a Gaussian distribution $\mathcal{N}(0, I)$, where $I$ represents the identity matrix. Similarly, the dataset U comprises 10,000 images, with each pixel value sampled from a uniform distribution $\mathcal{U}(-1, 1)$. The dataset OODom consists of 10,000 images, with each pixel value sampled from a uniform distribution $\mathcal{U}(-10, 10)$. The dataset Constant is obtained by selecting a single pixel value from a uniform distribution $\mathcal{U}(-1, 1)$ and expanding it to create 10,000 images based on the size of the original image. These created datasets were used directly without projection into the image domain. Tab. 3 presents the statistics of the non-natural OOD datasets we used.

Table 3: Statistics of non-natural OOD datasets

|  | Mean | Std. | Max | Min |
|---|---|---|---|---|
| N | 0.0004 | 1 | 6.052 | -5.807 |
| U | 0.0 | 0.5774 | 1 | -1 |
| OODom | -0.0002 | 5.7726 | 10 | -10 |
| Constant | 0.0028 | 0.5766 | 1 | -1 |

**OOD Datasets in Tab. 2**   In comparative experiments, we utilized the following six OOD datasets: SVHN, Texture, iSUN, LSUN, LSUN_R, and Places365. These six datasets were chosen for their popularity as evaluation benchmarks.

## G   EXPLORATION

### G.1   ABLATION STUDY

We present ablation studies for the WRN-28-10 model trained on CIFAR-10 as the ID dataset. The OOD dataset, consisting of CIFAR-100, CIFAR-10 Interp., N, U, OODom, and Constant, is evaluated based on FPR95 and AUROC metrics. We use the average of the AUROC metric as guidance for the design choices.

**ECH Modules**  Tab. 4 provides an overview of the ablation study conducted on the three modules employed in the ECH. We use the negative marginal energy and the maximum conditional likelihood as detection criteria. When excluding the attenuation factor, the performance decreases by 2.22% and 2.68%, respectively. Similarly, excluding the kernel results in a performance decrease of 1.59% and 1.04%. Furthermore, when these factors are excluded, the performance decreases by 4.35% and 5.01%. Thus, in the proposed ECH, all modules are essential, each playing a complementary role. Furthermore, based on FPR, the module without the kernel exhibits superior performance.

Table 4: ECH modules

| | | CIFAR-100 | | CIFAR-10 Interp. | | N | | U | | OODom | | Constant | | AVG | |
|---|---|---|---|---|---|---|---|---|---|---|---|---|---|---|---|
| | | FPR | AUROC | FPR | AUROC | FPR | AUROC | FPR | AUROC | FPR | AUROC | FPR | AUROC | FPR | AUC |
| $-E(\mathbf{x})$ | $\omega\cos(\theta_{(\mathbf{w}_y,\mathbf{z})})$ | 57 | 87.94 | 73.7 | 78.33 | 11.1 | 98.12 | 0 | 99.93 | 53.1 | 94.95 | 53.1 | 94.95 | 41.33 | 92.37 |
| | w/o $\Phi_y(\mathbf{z})$ | **16.9** | 97.49 | **64.2** | 73.33 | 0 | **99.99** | 0 | 99.99 | 0 | **100** | 0 | **100** | **13.52** | 95.13 |
| | w/o $\Psi(s_c(\mathbf{z}))$ | 51.3 | 89.54 | 73.3 | 77.94 | 0 | 99.76 | 0.1 | 99.78 | 0 | **100** | 0 | **100** | 20.78 | 94.50 |
| | ECH | 21.5 | 97.13 | 66.7 | **83.29** | 0.3 | 99.9 | 0 | **100** | 0 | 99.99 | 0 | 99.99 | 14.75 | **96.72** |
| $\max_y p(y\vert\mathbf{x})$ | $\omega\cos(\theta_{(\mathbf{w}_y,\mathbf{z})})$ | 61.5 | 88.55 | 76.1 | 78.8 | 39.2 | 95.41 | 0.6 | 99.43 | 89.3 | 92 | 89.3 | 92 | 59.33 | 91.03 |
| | w/o $\Phi_y(\mathbf{z})$ | **33.9** | **95.21** | **70.3** | 74.76 | 0 | **100** | 0 | **100** | 0 | **100** | 0 | **100** | **17.37** | 95.00 |
| | w/o $\Psi(s_c(\mathbf{z}))$ | 60.6 | 88.81 | 75.7 | 78.88 | 3 | 98.31 | 1.1 | 98.68 | 0 | 97.75 | 0 | 97.75 | 23.40 | 93.36 |
| | ECH | 39.5 | 94.81 | 72.5 | **82.32** | 6.3 | 99.1 | 0 | **100** | 0 | **100** | 0 | **100** | 19.72 | **96.04** |

Table 5: Kernel functions

| | | CIFAR-100 | | CIFAR-10 Interp. | | N | | U | | OODom | | Constant | | AVG | |
|---|---|---|---|---|---|---|---|---|---|---|---|---|---|---|---|
| | | FPR | AUROC | FPR | AUROC | FPR | AUROC | FPR | AUROC | FPR | AUROC | FPR | AUROC | FPR | AUC |
| $-E(\mathbf{x})$ | recip. | **21.5** | **97.13** | **66.7** | **83.29** | 0.3 | 99.9 | 0 | **100** | 0 | 99.99 | 0 | 99.99 | **14.75** | **96.72** |
| | Gaussian | 26.5 | 96.29 | 69.4 | 77.56 | 0 | **99.99** | 0 | 99.99 | 0 | 99.99 | 0 | 99.99 | 15.98 | 95.64 |
| $\max_y p(y\vert\mathbf{x})$ | recipe. | **39.5** | **94.81** | **72.5** | **82.32** | 6.3 | 99.1 | 0 | **100** | 0 | **100** | 0 | **100** | **19.72** | **96.04** |
| | Gaussian | 44.9 | 93.91 | 74.2 | 78.45 | 0 | **99.99** | 0 | 100 | 0 | 100 | 0 | 100 | 19.85 | 95.39 |

**Kernel Function**  Tab. 5 presents a comparative analysis of the kernel functions utilized in the ECH kernel. The evaluation is based on calculating the average AUROC values over the OOD dataset. The performance demonstrates 1.08% and 0.65% increase when utilizing the reciprocal kernel; this observation holds across all OOD datasets.

**Hyperparameters**  Tab. 6 shows the ablation study on the hyperparameters employed in the ECH. The feature dimension in the projection space is set to 64, and the projection map $\phi$ consists of one hidden layer with SeLU [37] activation function, chosen due to its performance advantage over other alternatives.

Table 6: ECH Hyperparemeters

| | | CIFAR-100 | | CIFAR-10 Interp. | | N | | U | | OODom | | Constant | | AVG | |
|---|---|---|---|---|---|---|---|---|---|---|---|---|---|---|---|
| | | FPR | AUROC | FPR | AUROC | FPR | AUROC | FPR | AUROC | FPR | AUROC | FPR | AUROC | FPR | AUC |
| $-E(\mathbf{x})$ | # of hidden layers = 0 | **16.8** | 97.37 | **64.5** | 74 | 0 | **100** | 0 | 100 | 0 | **100** | 0 | **100** | **13.55** | 95.23 |
| | # of hidden layers = 1 | 21.5 | 97.13 | 66.7 | **83.29** | 0.3 | 99.9 | 0 | **100** | 0 | 99.99 | 0 | 99.99 | 14.75 | **96.72** |
| | # of hidden layers = 2 | 17 | **97.56** | 66.6 | 70.68 | 0 | **100** | 0 | 100 | 0 | **100** | 0 | **100** | 13.93 | 94.71 |
| $\max_y p(y\vert\mathbf{x})$ | # of hidden layers = 0 | 34.7 | 94.98 | **72.2** | 75.18 | 0 | **100** | 0 | 100 | 0 | **100** | 0 | **100** | **17.82** | 95.03 |
| | # of hidden layers = 1 | 39.5 | 94.81 | 72.5 | **82.32** | 6.3 | 99.1 | 0 | **100** | 0 | **100** | 0 | **100** | 19.72 | **96.04** |
| | # of hidden layers = 2 | **34.6** | **95.35** | 72.9 | 73.02 | 0 | **100** | 0 | 100 | 0 | **100** | 0 | **100** | 17.92 | 94.73 |
| $-E(\mathbf{x})$ | feat. dim. = 32 | 17.4 | 97.55 | **63.5** | 75.09 | 0 | **99.99** | 0 | 99.99 | 0 | 99.99 | 0 | 99.99 | **13.48** | 95.43 |
| | feat. dim. = 64 | 21.5 | 97.13 | 66.7 | **83.29** | 0.3 | 99.9 | 0 | **100** | 0 | 99.99 | 0 | **99.99** | 14.75 | **96.72** |
| | feat. dim. = 128 | **17.3** | **97.64** | 65.5 | 73.06 | 0 | **100** | 0 | 100 | 0 | **100** | 0 | 100 | 13.80 | 95.12 |
| $\max_y p(y\vert\mathbf{x})$ | feat. dim. = 32 | 37.1 | 95.2 | 71.8 | 76.24 | 0 | **100** | 0 | 100 | 0 | 99.99 | 0 | 99.99 | 18.15 | 95.24 |
| | feat. dim. = 64 | 39.5 | 94.81 | 72.5 | **82.32** | 6.3 | 99.1 | 0 | **100** | 0 | **100** | 0 | **100** | 19.72 | **96.04** |
| | feat. dim. = 128 | **35.8** | **95.3** | **71.7** | 74.69 | 0 | **100** | 0 | 100 | 0 | **100** | 0 | **100** | **17.92** | 95.00 |
| $-E(\mathbf{x})$ | SeLU [37] | 21.5 | 97.13 | 66.7 | **83.29** | 0.3 | 99.9 | 0 | **100** | 0 | 99.99 | 0 | 99.99 | 14.75 | **96.72** |
| | Leaky ReLU | **14.3** | **97.99** | **60.6** | 80.9 | 0 | **99.99** | 0 | 99.99 | 0 | 99.99 | 0 | 99.99 | **12.48** | 96.48 |
| | tanh | 29 | 95.65 | 67.5 | 78.77 | 0 | 99.99 | 0 | 99.96 | 0 | 99.99 | 0 | 99.99 | 16.08 | 95.73 |
| $\max_y p(y\vert\mathbf{x})$ | SeLU [37] | 39.5 | 94.81 | 72.5 | **82.32** | 6.3 | 99.1 | 0 | **100** | 0 | **100** | 0 | **100** | 19.72 | **96.04** |
| | Leaky ReLU | **32.2** | **95.79** | 69.4 | 80.32 | 0 | 99.94 | 0 | 100 | 0 | **100** | 0 | **100** | **16.93** | 96.01 |
| | tanh | 45.8 | 93.4 | 72.9 | 79.17 | 0 | **99.99** | 3 | 99.49 | 0 | 99.99 | 0 | 99.99 | 20.28 | 95.34 |

## G.2  ARCHITECTURAL DEPENDENCY

We used the WRN-28-10 for the analysis tasks. We used the WRN-28-10, ResNet18, and ResNet34 models trained **five times** and compared in the OOD detection tasks. The results indicate that the ECH outperforms LinH over all models and datasets. Notably, the performance is maximized when using the ResNet34 model. Therefore, the ResNet34 model was additionally trained to obtain **ten results** before comparing it with baselines.

Table 7: Architectural Dependency

| | | | | SVHN[55] | | Texture[6] | | iSUN[71] | | LSUN[74] | | Places365[79] | | LSUN_R[74] | | AVG | |
|---|---|---|---|---|---|---|---|---|---|---|---|---|---|---|---|---|---|
| | | | ACC | FPR | AUROC | FPR | AUROC | FPR | AUROC | FPR | AUROC | FPR | AUROC | FPR | AUROC | FPR | AUROC |
| WRN-28-10 | $-E(\mathbf{x})$ | LinH | 95.34 ± 0.07 | 41.66 ± 15.4 | 89.89 ± 7.7 | 51.1 ± | 86.31 ± 7.5 | 19.68 ± 5.4 | 96.35 ± 1.1 | 34.44 ± 11.2 | 92.71 ± 3.4 | 40.56 ± 3.1 | 89.94 ± 1.3 | 17.76 ± 4.6 | 95.64 ± 0.9 | 34.2 ± 7.8 | 91.81 ± 3.2 |
| | | ECH | 95.13 ± 0.07 | **10.64** ± 12.5 | **97.87** ± 3 | **20.28** ± 1.6 | **95.58** ± 1.4 | **7.44** ± 5.8 | **98.89** ± 0.96 | **18.66** ± 4.5 | **97.4** ± 0.6 | **28.38** ± 4.9 | **94.72** ± 1.9 | **5.84** ± 3.7 | **99.18** ± 0.5 | **15.21** ± 4.5 | **97.27** ± 1.8 |
| | $\max_y p(y\|\mathbf{x})$ | LinH | 95.34 ± 0.07 | 51.25 ± 8.7 | 90.12 ± 4.7 | 55.33 ± 3.1 | 88.32 ± 2.1 | 36.83 ± 3.8 | 94.59 ± 0.6 | 46.1 ± 5.5 | 92.01 ± 1.8 | 52.13 ± 1.9 | 89.54 ± 0.8 | 35.78 ± 3.8 | 94.78 ± 0.6 | 46.24 ± 5.5 | 91.56 ± 1.4 |
| | | ECH | 95.13 ± 0.07 | **19.04** ± 11.8 | **97.26** ± 2.1 | **30.8** ± 1.4 | **94.29** ± 1.2 | **15.66** ± 6.8 | **97.79** ± 1 | **31.12** ± 5.2 | **95.97** ± 0.6 | **40.58** ± 4.1 | **93.19** ± 1.5 | **14.1** ± 5.4 | **98.02** ± 0.7 | **25.22** ± 5.8 | **96.09** ± 1.2 |
| Resnet34 | $-E(\mathbf{x})$ | LinH | 95.62 ± 0.08 | 15.02 ± 5.2 | 95.92 ± 1.1 | 34.64 ± 2.7 | 90.98 ± 0.9 | 13.72 ± 2.9 | 97.54 ± 0.5 | 25.48 ± 3.0 | 94.29 ± 0.6 | 39.39 ± 1.3 | 87.97 ± 1.0 | 13.65 ± 2.6 | 97.46 ± 0.6 | 23.65 ± 3.0 | 94.03 ± 0.8 |
| | | ECH | 95.46 ± 0.06 | **5.51** ± 1.16 | **99.14** ± 0.14 | **22.87** ± 2.51 | **95.9** ± 0.36 | **4.32** ± 1.65 | **99.22** ± 0.17 | **10.95** ± 4.58 | **98.58** ± 0.44 | **32.89** ± 4.29 | **94.57** ± 1.03 | **4.35** ± 1.39 | **99.2** ± 0.16 | **13.48** ± 2.6 | **97.77** ± 0.38 |
| | $\max_y p(y\|\mathbf{x})$ | LinH | 95.61 ± 0.06 | 27.52 ± 6.5 | 94.79 ± 1.0 | 46.25 ± 2.8 | 90.86 ± 0.7 | 29.27 ± 4.0 | 95.86 ± 0.6 | 41.63 ± 5.4 | 92.8 ± 0.6 | 51.91 ± 1.3 | 87.54 ± 0.7 | 29.64 ± 4.0 | 95.76 ± 0.5 | 37.7 ± 3.9 | 92.94 ± 0.7 |
| | | ECH | 95.46 ± 0.06 | **7.55** ± 2.1 | **98.92** ± 0.26 | **25.81** ± 2.52 | **95.73** ± 0.32 | **6.55** ± 2.17 | **98.87** ± 0.22 | **14.49** ± 6.16 | **97.65** ± 1.54 | **37.77** ± 4.16 | **94.2** ± 0.89 | **6.66** ± 2.19 | **98.81** ± 0.25 | **17.36** ± 3.22 | **97.36** ± 0.58 |
| Resnet18 | $-E(\mathbf{x})$ | LinH | 95.57 ± 0.09 | 20.82 ± 7.6 | 95.69 ± 1.2 | 35.4 ± 1.7 | 91.43 ± 0.9 | 21.08 ± 6.0 | 95.58 ± 1.6 | 23.84 ± 2.9 | 94.66 ± 0.8 | 41.28 ± 1.6 | 88.68 ± 0.6 | 21.56 ± 4.2 | 95.65 ± 0.9 | 27.33 ± 4.0 | 93.62 ± 1.0 |
| | | ECH | 95.26 ± 0.1 | **9.04** ± 2.9 | **98.78** ± 0.3 | **22.4** ± 2.3 | **96.142** ± 0.8 | **4.54** ± 1.7 | **99.26** ± 0.2 | **15.6** ± 5.6 | **98.09** ± 0.6 | **33.62** ± 1.5 | **94.7** ± 0.5 | **5.08** ± 1.3 | **99.19** ± 0.2 | **15.05** ± 2.6 | **97.69** ± 0.4 |
| | $\max_y p(y\|\mathbf{x})$ | LinH | 95.57 ± 0.07 | 39.5 ± 7.6 | 94.22 ± 0.9 | 47.82 ± 1.7 | 91.22 ± 0.5 | 37.92 ± 4.8 | 94.16 ± 1.1 | 41.46 ± 1.9 | 93.29 ± 0.4 | 54 ± 1.3 | 88.23 ± 0.5 | 38.22 ± 3.7 | 94.19 ± 0.6 | 43.15 ± 3.5 | 92.55 ± 0.7 |
| | | ECH | 95.29 ± 0.1 | **12.14** ± 3.2 | **98.48** ± 0.4 | **26.1** ± 2.1 | **95.87** ± 0.8 | **7.04** ± 2.9 | **98.88** ± 0.3 | **19.72** ± 5.4 | **97.71** ± 0.6 | **39.6** ± 2.4 | **94.31** ± 0.5 | **7.92** ± 2.2 | **98.75** ± 0.2 | **18.75** ± 3.1 | **97.33** ± 0.5 |

# H ADDITIONAL RESULTS

We present supplementary experimental results in this section.

## H.1 SCALE FACTORS FROM METACOGNITION MODULE IN ECH

The statistics of scale factors of the metacognition module for the ID test set are 0.9817 (mean) and 0.1366 (std.). Conversely, for OOD, the computed scale factors are as follows. SVHN: 0.2447 (mean), 0.2891 (std.); Texture: 0.4719 (mean), 0.3758 (std.); iSUN: 0.3177 (mean), 0.2749 (std.); LSUN: 0.3861 (mean), 0.3254 (std.); LSUN_R: 0.3356 (mean), 0.2744 (std.); and Places365: 0.7339 (mean), 0.2565 (std.). These results collectively attest to the relatively precise predictive efficacy of pre-marginal energy for the ID samples.

## H.2 QUANTITATIVE RESULTS OF OOD CALIBRATION

Table 8: AUC Error of Fig. 1

| | AUCE | |
|---|---|---|
| | $(\max_y p(y\|\mathbf{x}))$ | $-H(P(y\|\mathbf{x}))$ |
| Human | – | – |
| LinH | 0.2935 | 0.9227 |
| CosH | 0.2553 | 0.6657 |
| GPH | 0.2501 | 0.5969 |
| **ECH** | **0.1644** | **0.4954** |

The table presents quantitative results regarding AUCE, as depicted in Fig. 1. Compared to LinH based on maximum likelihood (a prevalent head), ECH demonstrates a 44.0% improvement. Additionally, compared to LinH based on entropy, ECH exhibits a 46.3% improvement. On average, ECH shows an improvement of 45.1%.

## H.3 EMPIRICAL OBSERVATION OF ID AND OOD DATASETS

In Fig. 17(a), lines of ID and OOD have little or no overlap in one standard deviation range at a high confidence range, making it relatively easy to distinguish. In addition, despite the difference in cosine similarity, the conditional probabilities (vertical axis) appear in the same bin, suggesting that LinH can have similar conditional probabilities at different negative marginal energies. In Fig. 17(b), the norm of ID appears smaller than that of OOD at a high confidence range, and ID and OOD can be distinguished within the range of one standard deviation.

## H.4 UNCERTAINTY CALIBRATION FOR DISTANT SAMPLES

Tab. 9 presents the uncertainty calibration for distant samples numerically. As elucidated in Appendix F.2, it shows the average and standard deviation of the conditional likelihood entropy for grid

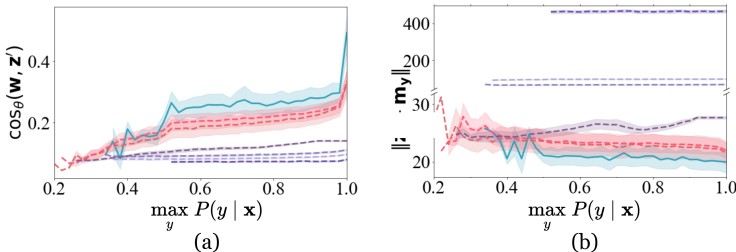

(a)                                    (b)

Figure 11: Empirical observation of the properties of latent vectors and logits for the ID and OOD test sets. We use the WRN-28-10 (LinH) model trained on the CIFAR-10 (ID) dataset. The horizontal axis represents $\max_y P(y|\mathbf{x})$, a solid line represents the mean value of each bin, the shaded region represents one standard deviation area, and each legend represents the ID and OOD dataset (Appendix C). The maximum conditional likelihood is divided into 20 bins. (a) The vertical axis represents the cosine similarity between the latent vector and the weights of the $y^*$ class. (b) The vertical axis represents the norm of the deviation between the latent vector and the average vector of the latent vector for each ID class.

inputs in random directions from randomly selected ID testset inputs. It is challenging to determine a clear advantage when the uncertainty is high or low near the ID input. However, as one moves further away, higher uncertainty in the out-of-domain (OOD) input can be anticipated, and ECH effectively diplays this trend.

Table 9: Uncertainty Calibration for Distant Samples

| input grid scale: | $10^0$ | $10^1$ | $10^2$ |
|---|---|---|---|
| LinH | 0.081 | 0.366 | 0.212 |
| | $\pm$ 0.007 | $\pm$ 0.101 | $\pm$ 0.292 |
| CosH | **0.216** | 0.554 | 1.138 |
| | $\pm$ 0.004 | $\pm$ 0.167 | $\pm$ 0.392 |
| GPH | 0.159 | 0.309 | 1.54 |
| | $\pm$ 0.001 | $\pm$ 0.088 | $\pm$ 0.451 |
| ECH | 0.139 | **0.878** | **2.277** |
| | $\pm$ 0.007 | $\pm$ 0.41 | $\pm$ 0.162 |

## H.5 ADVERSARIAL DETECTION

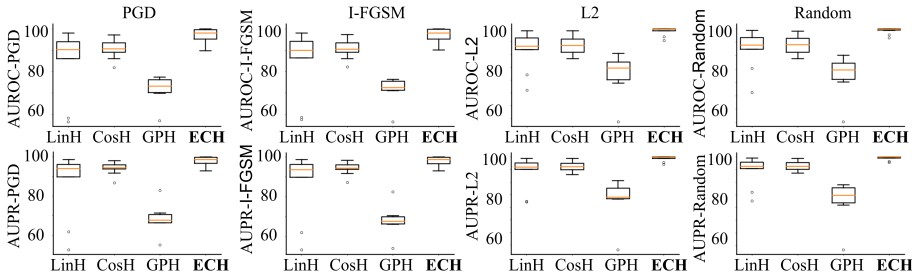

Figure 12: Box plots of each head type's adversarial detection

We conducted ten times of training and evaluation for each head in adversarial detection. The results are presented in the box plot shown in Fig. 12, demonstrating that the ECH has clear advantages over other heads across all types of attacks, evaluation metrics, and OOD detection metrics. Furthermore, we present the ID for each head and the histogram of adversarial samples in Figs. 13 and 14. This additional information is provided to enhance model performance evaluation in adversarial detection.

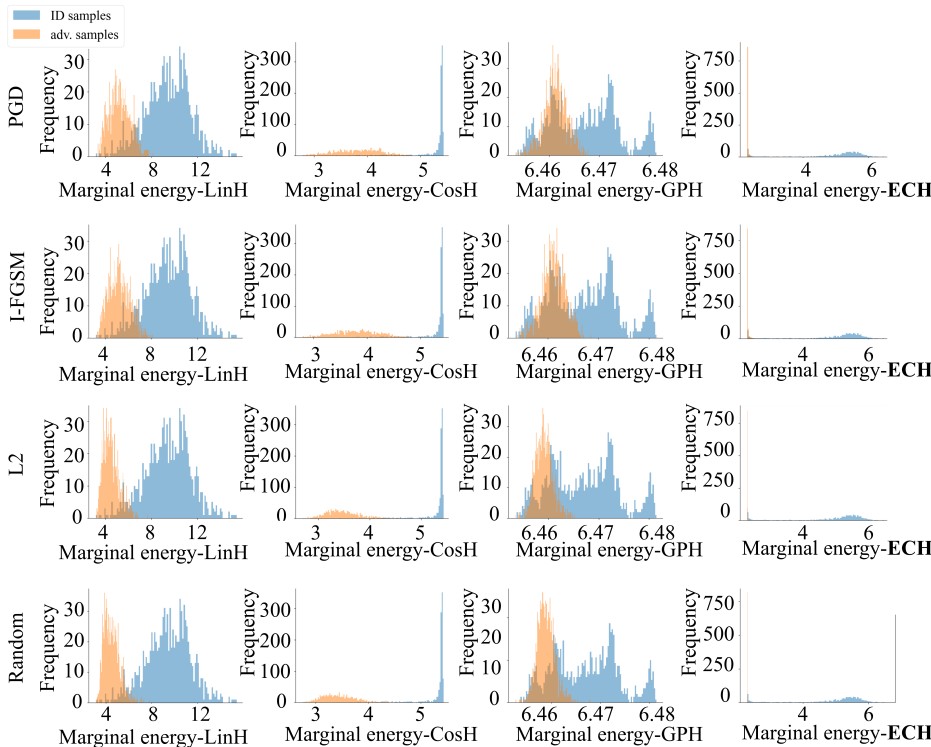

Figure 13: Histograms illustrating the distribution of marginal energies for both ID and adversarial samples generated through PGD, I-FGSM, L2, and random attacks (20 steps; $\epsilon$=8/255) across different heads

## H.6 OOD DETECTION

### H.6.1 COMPARISON WITH RELATED WORKS

JEM [22] is a hybrid model considering a classifier as an energy-based model (EBM). It has demonstrated its versatility and effectiveness across diverse tasks. Drawing inspiration from JEM, ECH extracts the marginal energy from a classifier and leverages it for uncertainty calibration. Meanwhile, it is worth noting that JEM is acknowledged for its inherent difficulty in successful training. Despite our efforts, we were unable to achieve acceptable results using public code written by the authors. Consequently, we opted to utilize the reported figures from [22] for comparative evaluation with the ECH, particularly concerning OOD detection. As in Tab. 10, ECH exhibits a superior performance across most results while avoiding the additional computational overhead associated with data sampling that JEM has.

Table 10: Comparative results for OOD detection: ECH vs. JEM

|  |  | AUROC | | | |
|---|---|---|---|---|---|
|  |  | SVHN | Interp | CIFAR100 | CelebA |
| JEM [22] | $(-E(\mathbf{x}))$ | 67 | 65 | 67 | 75 |
|  | $(\max_y p(y\|\mathbf{x}))$ | 89 | **75** | 87 | 79 |
| ECH | $(-E(\mathbf{x}))$ | **99.14** | 67.19 | **93.94** | **84.49** |
|  | $(\max_y p(y\|\mathbf{x}))$ | 98.92 | 66.93 | 93.36 | 82.33 |

A seminal work, SNGP [46], integrates weight normalization and a Gaussian process (GP) layer to model uncertainty and demonstrates its usefulness across various tasks, including out-of-distribution (OOD) detection. This research aligns with our approach to in-network uncertainty modeling. We compare ECH and SNGP performance based on the reported figures in [46].

As in Tab. 11, ECH exhibits a comparable performance.

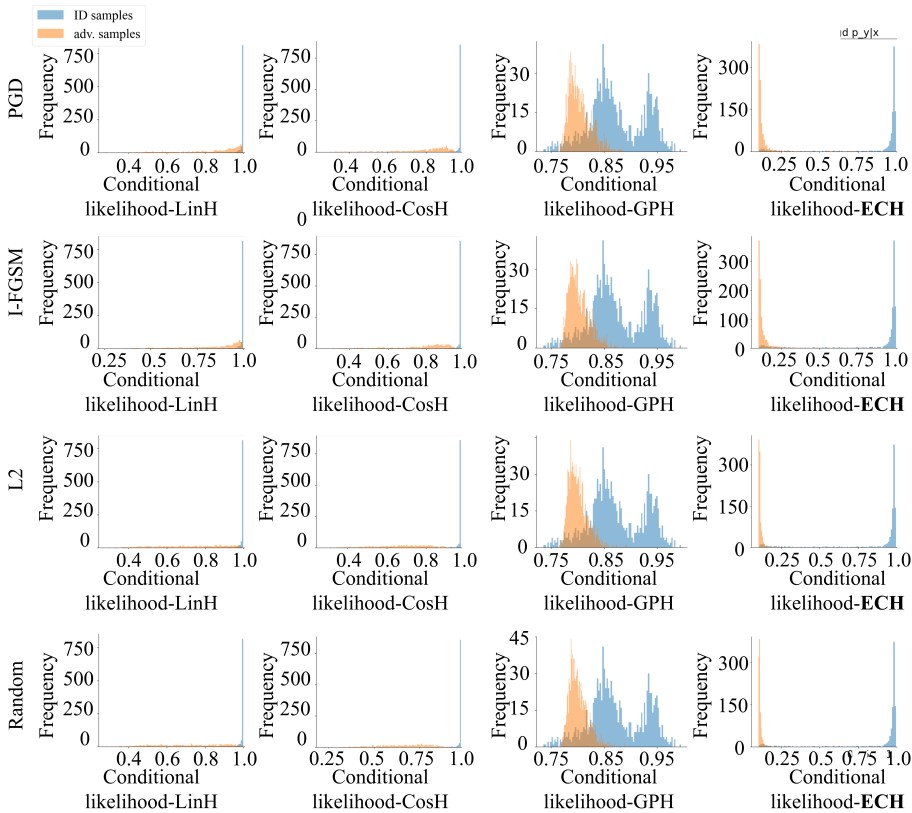

Figure 14: Histograms illustrating the distribution of conditional likelihood for both ID and adversarial samples generated through PGD, I-FGSM, L2, and random attacks (20 steps; $\epsilon$=8/255) across different heads.

Table 11: Comparative results for OOD detection: ECH vs. SNGP

|  | AUPR↑ | |
|  | SVHN | CIFAR100 |
| --- | --- | --- |
| SNGP [46] | **0.990** ± 0.01 | 0.905 ± 0.01 |
| ECH | 0.983 ± 0.502 | **0.951** ± 0.632 |

### H.6.2 STATISTICAL SIGNIFICANCE

We aim to validate the statistical significance of various heads in Tab. 12 and baselines in Tab. 13, each independently trained **ten times** using the ID dataset. For each dataset employed in OOD detection, we conduct the Mann-Whitney U test on the AUROC values of the ten models trained of each head and baseline. The Mann-Whitney U test, a non-parametric test known as the Wilcoxon rank-sum test, compares the metrics of OOD detection between LinH and ECH. This test assesses whether the metrics from two groups of samples are drawn from the same distribution. Unlike the $t$-test, the Mann-Whitney U test does not assume a Gaussian distribution for the metrics of the two groups.

In Tab. 12, the $p$-values obtained from the test conducted between pairs of baseline head and ECH for each model and OOD dataset are displayed in $10^{-3}$ scale. All resulting p-values are lower than the commonly used significance level of 0.01. Consequently, we reject the hypothesis that the results are sampled from the same distribution. Thus, we conclude that all metrics of ECH are significantly better than the other head.

In Tab. 13, we present the $p$-values derived from the test conducted between pairs of the baseline and ECH for each metric and OOD dataset. Among the 108 comparison cases, 100 resulting p-values are lower than the commonly used significance level of 0.01. Therefore, we reject the hypothesis that the results are sampled from the same distribution. Then, the ECH shows significantly better performance in 96 cases. Consequently, ECH significantly outperforms the baselines across most benchmarks and metrics.

Table 12: p-values of the Mann-Whitney U test between pairs of the heads and ECH for adversarial detection. A parenthesis indicates the case where the $p$-value exceeds 0.01. A square bracket represents that the mean of the performance distribution of a baseline is better than the ECH.

|  |  | PGD [49] | I-FGSM [39] | L2 [39] | Random |
|---|---|---|---|---|---|
| LinH | AUROC | $5.056 \times 10^{-3}$ | $3.116 \times 10^{-3}$ | $7.18 \times 10^{-4}$ | $4.69 \times 10^{-4}$ |
|  | AUPR | $5.039 \times 10^{-3}$ | $3.982 \times 10^{-3}$ | $9.50 \times 10^{-4}$ | $5.39 \times 10^{-4}$ |
| CosH | AUROC | $3.642 \times 10^{-3}$ | $3.642 \times 10^{-3}$ | $5.02 \times 10^{-4}$ | $2.19 \times 10^{-4}$ |
|  | AUPR | $7.009 \times 10^{-3}$ | $8.629 \times 10^{-3}$ | $5.02 \times 10^{-4}$ | $2.90 \times 10^{-4}$ |
| GPH | AUROC | $1.347 \times 10^{-3}$ | $1.347 \times 10^{-3}$ | $1.335 \times 10^{-3}$ | $1.335 \times 10^{-3}$ |
|  | AUPR | $1.347 \times 10^{-3}$ | $1.347 \times 10^{-3}$ | $1.335 \times 10^{-3}$ | $1.335 \times 10^{-3}$ |

Table 13: $p$-values of the Mann-Whitney U test between pairs of the baseline methods and ECH for OOD detection

|  |  | SVHN[55] | Texture[6] | iSUN[71] | LSUN[74] | Places365[79] | LSUN_R[74] |
|---|---|---|---|---|---|---|---|
| LinH | AUROC | $9.032 \times 10^{-5}$ | $9.134 \times 10^{-5}$ | $9.134 \times 10^{-5}$ | $1.649 \times 10^{-4}$ | $9.134 \times 10^{-5}$ | $9.134 \times 10^{-5}$ |
|  | FPR | $1.224 \times 10^{-4}$ | $9.134 \times 10^{-5}$ | $9.083 \times 10^{-5}$ | $2.293 \times 10^{-3}$ | $8.831 \times 10^{-5}$ | $9.134 \times 10^{-5}$ |
|  | AUPR | $9.134 \times 10^{-5}$ | $9.134 \times 10^{-5}$ | $9.083 \times 10^{-5}$ | $1.231 \times 10^{-4}$ | $9.134 \times 10^{-5}$ | $9.083 \times 10^{-5}$ |
| CosH | AUROC | $1.381 \times 10^{-4}$ | $1.399 \times 10^{-4}$ | $1.399 \times 10^{-4}$ | $2.601 \times 10^{-4}$ | $1.399 \times 10^{-4}$ | $1.390 \times 10^{-4}$ |
|  | FPR | $1.399 \times 10^{-4}$ | $1.399 \times 10^{-4}$ | $1.399 \times 10^{-4}$ | $1.913 \times 10^{-4}$ | $1.390 \times 10^{-4}$ | $1.399 \times 10^{-4}$ |
|  | AUPR | $1.399 \times 10^{-4}$ | $(1.371 \times 10^{-2})$ | $1.390 \times 10^{-4}$ | $1.440 \times 10^{-4}$ | $1.399 \times 10^{-4}$ | $1.399 \times 10^{-4}$ |
| GPH | AUROC | $3.745 \times 10^{-4}$ | $3.802 \times 10^{-4}$ | $3.802 \times 10^{-4}$ | $3.802 \times 10^{-4}$ | $3.802 \times 10^{-4}$ | $3.802 \times 10^{-4}$ |
|  | FPR | $3.689 \times 10^{-4}$ | $3.802 \times 10^{-4}$ | $3.802 \times 10^{-4}$ | $3.802 \times 10^{-4}$ | $3.773 \times 10^{-4}$ | $3.802 \times 10^{-4}$ |
|  | AUPR | $1.056 \times 10^{-3}$ | $3.802 \times 10^{-4}$ | $3.773 \times 10^{-4}$ | $3.802 \times 10^{-4}$ | $3.802 \times 10^{-4}$ | $3.802 \times 10^{-4}$ |
| MSP[29] | AUROC | $8.981 \times 10^{-5}$ | $9.134 \times 10^{-5}$ | $9.134 \times 10^{-5}$ | $9.134 \times 10^{-5}$ | $9.134 \times 10^{-5}$ | $9.134 \times 10^{-5}$ |
|  | FPR | $9.134 \times 10^{-5}$ | $9.134 \times 10^{-5}$ | $9.134 \times 10^{-5}$ | $9.134 \times 10^{-5}$ | $9.032 \times 10^{-5}$ | $9.083 \times 10^{-5}$ |
|  | AUPR | $9.134 \times 10^{-5}$ | $9.134 \times 10^{-5}$ | $9.083 \times 10^{-5}$ | $9.134 \times 10^{-5}$ | $9.134 \times 10^{-5}$ | $9.134 \times 10^{-5}$ |
| ODIN[44] | AUROC | $9.032 \times 10^{-5}$ | $9.032 \times 10^{-5}$ | $9.032 \times 10^{-5}$ | $(1.557 \times 10^{-2})$ | $9.134 \times 10^{-5}$ | $8.881 \times 10^{-5}$ |
|  | FPR | $9.134 \times 10^{-5}$ | $9.134 \times 10^{-5}$ | $9.032 \times 10^{-5}$ | $(4.849 \times 10^{-1})$ | $9.032 \times 10^{-5}$ | $2.198 \times 10^{-4}$ |
|  | AUPR | $9.134 \times 10^{-5}$ | $9.134 \times 10^{-5}$ | $9.032 \times 10^{-5}$ | $8.629 \times 10^{-3}$ | $9.083 \times 10^{-5}$ | $9.083 \times 10^{-5}$ |
| GODIN[31] | AUROC | $9.032 \times 10^{-5}$ | $9.134 \times 10^{-5}$ | $9.134 \times 10^{-5}$ | $5.649 \times 10^{-3}$ | $9.134 \times 10^{-5}$ | $9.134 \times 10^{-5}$ |
|  | FPR | $9.134 \times 10^{-5}$ | $9.134 \times 10^{-5}$ | $9.134 \times 10^{-5}$ | $(1.923 \times 10^{-1})$ | $9.083 \times 10^{-5}$ | $9.134 \times 10^{-5}$ |
|  | AUPR | $9.134 \times 10^{-5}$ | $1.231 \times 10^{-4}$ | $9.083 \times 10^{-5}$ | $5.649 \times 10^{-3}$ | $9.134 \times 10^{-5}$ | $9.134 \times 10^{-5}$ |
| ReAct[65] | AUROC | $9.032 \times 10^{-5}$ | $9.134 \times 10^{-5}$ | $9.134 \times 10^{-5}$ | $3.843 \times 10^{-4}$ | $9.134 \times 10^{-5}$ | $9.134 \times 10^{-5}$ |
|  | FPR | $9.134 \times 10^{-5}$ | $9.134 \times 10^{-5}$ | $9.134 \times 10^{-5}$ | $4.554 \times 10^{-3}$ | $9.083 \times 10^{-5}$ | $9.134 \times 10^{-5}$ |
|  | AUPR | $9.134 \times 10^{-5}$ | $9.134 \times 10^{-5}$ | $9.083 \times 10^{-5}$ | $1.418 \times 10^{-4}$ | $9.134 \times 10^{-5}$ | $9.134 \times 10^{-5}$ |
| LogitNorm[70] | AUROC | $1.217 \times 10^{-4}$ | $9.083 \times 10^{-5}$ | $1.231 \times 10^{-4}$ | $5.040 \times 10^{-4}$ | $1.231 \times 10^{-4}$ | $2.198 \times 10^{-4}$ |
|  | FPR | $9.134 \times 10^{-5}$ | $9.134 \times 10^{-5}$ | $2.198 \times 10^{-4}$ | $9.134 \times 10^{-5}$ | $(2.853 \times 10^{-1})$ | $3.843 \times 10^{-4}$ |
|  | AUPR | $[9.083 \times 10^{-5}]$ | $[9.083 \times 10^{-5}]$ | $(3.466 \times 10^{-2})$ | $[8.831 \times 10^{-5}]$ | $[9.083 \times 10^{-5}]$ | $[(4.697 \times 10^{-1})]$ |
| KNN[66] | AUROC | $9.032 \times 10^{-5}$ | $9.134 \times 10^{-5}$ | $9.134 \times 10^{-5}$ | $8.531 \times 10^{-4}$ | $9.083 \times 10^{-5}$ | $9.134 \times 10^{-5}$ |
|  | FPR | $9.134 \times 10^{-5}$ | $9.134 \times 10^{-5}$ | $9.134 \times 10^{-5}$ | $1.414 \times 10^{-3}$ | $9.083 \times 10^{-5}$ | $9.083 \times 10^{-5}$ |
|  | AUPR | $9.134 \times 10^{-5}$ | $(2.851 \times 10^{-1})$ | $9.083 \times 10^{-5}$ | $8.629 \times 10^{-3}$ | $9.083 \times 10^{-5}$ | $9.134 \times 10^{-5}$ |

### H.6.3 DETAILED EXPERIMENTAL RESULTS OF OOD DETECTION

We offer OOD detection results of ID (CIFAR-10 and SVHN) and OOD datasets (CIFAR-10/SVHN, Texture, iSUN, LSUN, Places365, LSUN_R, CIFAR-100, N, U, OODom, and Constant) with an additional evaluation metric (AUPR) in Tabs. 15 and 16.

### H.6.4 HISTOGRAM OF ID AND OOD SAMPLES

We present a histogram of ID and OOD samples for the LinH and ECH in Figs. 16 to 19, providing supplementary information to assess the model's performance in OOD detection.

### H.7 INFLUENCE OF LATENT NORM

We decompose Fig. 2 for each dataset to provide insights into how latent norm influences uncertainty and marginal energy across different datasets. It is important to note that, for the Constant dataset, a significant portion of the samples appears dense and does not exhibit due to the overlap along the axes when we use a fixed scale for every plot.

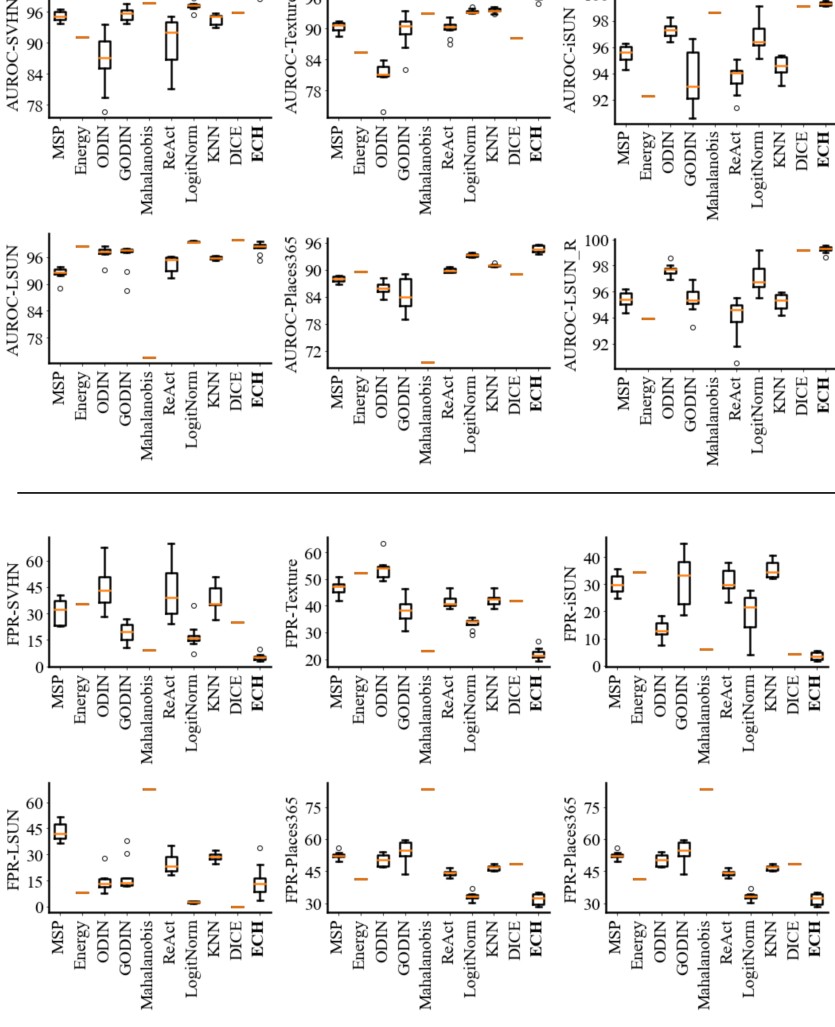

Figure 15: Box plots of OOD detection for each dataset and metric, comparing the set of baselines with ECH

Table 14: Comparative results of OOD detection with reported results

| | | SVHN[55] | | Texture[6] | | iSUN[71] | | LSUN[74] | | Places365[79] | | LSUN_R[74] | | AVG | | AVG w/o LSUN_R | |
|---|---|---|---|---|---|---|---|---|---|---|---|---|---|---|---|---|---|
| | | FPR | AUROC | FPR | AUROC | FPR | AUROC | FPR | AUROC | FPR | AUROC | FPR | AUROC | FPR | AUC | FPR | AUC |
| LinH | Mean | 17.67 | 96.45 | 34.47 | 90.67 | 14.58 | 97.29 | 27.91 | 93.81 | 40.12 | 88.54 | 14.86 | 97.17 | 24.94 | 93.99 | 29.27 | 92.58 |
| | Std. | ± 6.62 | ± 1.09 | ± 2.74 | ± 1.17 | ± 3.46 | ± 0.75 | ± 5.88 | ± 1.94 | ± 1.41 | ± 0.89 | ± 3.03 | ± 0.69 | ± 3.86 | ± 1.09 | ± 3.38 | ± 1.19 |
| CosH | Mean | 34.21 | 95.54 | 44.86 | 92.94 | 35.78 | 94.97 | 40.35 | 94.75 | 56.37 | 88.56 | 37.65 | 94.8 | 41.54 | 93.59 | 44.34 | 92.81 |
| | Std. | ± 12.62 | ± 1.28 | ± 2.18 | ± 0.61 | ± 5.75 | ± 0.91 | ± 6.18 | ± 0.69 | ± 1.84 | ± 0.56 | ± 5.25 | ± 0.69 | ± 5.64 | ± 0.79 | ± 3.99 | ± 0.7 |
| GPH | Mean | 99.95 | 94.3 | 99.08 | 85.56 | 99.01 | 84.66 | 99.7 | 91.35 | 98.9 | 82.37 | 98.97 | 84.71 | 99.27 | 87.16 | 99.18 | 85.99 |
| | Std. | ± 0.09 | ± 3.43 | ± 1.08 | ± 3.57 | ± 0.53 | ± 2.62 | ± 0.43 | ± 2.91 | ± 0.62 | ± 2.26 | ± 0.45 | ± 2.15 | ± 0.53 | ± 2.82 | ± 0.67 | ± 2.85 |
| MSP[29] | Mean | 31.02 | 95.236 | 46.54 | 90.311 | 30.17 | 95.53 | 43.09 | 92.503 | 52.2 | 88.027 | 30.8 | 95.40 | 38.97 | 92.84 | 43 | 91.59 |
| | Std. | ± 7.32 | ± 0.90 | ± 2.81 | ± 0.98 | ± 3.65 | ± 0.64 | ± 5.23 | ± 1.37 | ± 1.77 | ± 0.69 | ± 3.26 | ± 0.56 | ± 4.01 | ± 0.86 | ± 4.75 | ± 0.91 |
| | Reported | 59.66 | 91.25 | 66.45 | 88.50 | 54.57 | 92.12 | 62.46 | 88.64 | | | | | 57.67 | 90.86 | 57.27 | 90.78 |
| | Diff | 28.64 | 3.99 | 19.91 | 1.81 | 24.40 | 3.41 | 2.12 | 1.30 | 10.26 | 0.61 | | | 18.70 | 1.98 | 16.71 | 1.58 |
| ODIN[44] | Mean | 44.14 | 86.75 | 53.88 | 81.09 | 13.35 | 97.27 | 14.29 | 96.92 | 50.05 | 86.13 | 11.37 | 97.67 | 31.18 | 90.97 | 32.89 | 90.35 |
| | Std. | ± 12.54 | ± 5.52 | ± 4.01 | ± 2.75 | ± 3.26 | ± 0.58 | ± 5.51 | ± 1.45 | ± 2.84 | ± 1.49 | ± 2.62 | ± 0.48 | ± 5.13 | ± 2.05 | ± 5.23 | ± 2.35 |
| | Reported | 53.78 | 91.30 | 55.59 | 89.47 | 28.44 | 95.51 | 10.93 | 97.93 | 43.40 | 90.98 | | | 38.43 | 93.04 | 35.36 | 93.39 |
| | Diff | 9.64 | 4.55 | 1.71 | 8.38 | 15.09 | 1.76 | 3.36 | 1.01 | 6.65 | 4.85 | | | 7.25 | 2.07 | 6.77 | 1.57 |
| GODIN[31] | Mean | 19.41 | 95.52 | 38.33 | 89.69 | 31.48 | 93.68 | 17.65 | 96.17 | 54.00 | 84.49 | 23.73 | 95.48 | 30.77 | 92.51 | 35.37 | 91.01 |
| | Std. | ± 5.28 | ± 1.24 | ± 5.09 | ± 3.36 | ± 9.27 | ± 2.04 | ± 9.09 | ± 3.12 | ± 5.35 | ± 3.63 | ± 5.27 | ± 1.07 | ± 6.56 | ± 2.42 | ± 6.79 | ± 2.47 |
| | Reported | 18.72 | 96.10 | 33.58 | 92.20 | 30.02 | 94.02 | 11.52 | 97.12 | 55.25 | 85.50 | | | 29.82 | 92.99 | 32.04 | 92.37 |
| | Diff | 0.69 | 0.58 | 4.75 | 2.51 | 1.46 | 0.34 | 6.13 | 0.95 | 1.25 | 1.01 | | | 0.95 | 0.48 | 1.00 | 0.47 |
| ReAct[65] | Mean | 41.99 | 90.30 | 41.71 | 90.02 | 31.02 | 93.71 | 24.82 | 94.54 | 44.14 | 89.99 | 30.33 | 93.95 | 35.67 | 92.09 | 35.42 | 92.07 |
| | Std. | ± 14.87 | ± 5.01 | ± 2.56 | ± 1.53 | ± 4.66 | ± 1.09 | ± 5.76 | ± 1.82 | ± 1.61 | ± 0.48 | ± 5.95 | ± 1.60 | ± 5.91 | ± 1.93 | ± 5.29 | ± 2.18 |
| | Reported | 49.77 | 92.18 | 47.96 | 91.55 | 20.84 | 96.46 | 16.99 | 97.11 | 43.97 | 91.33 | 17.94 | 96.98 | 32.91 | 94.27 | 30.10 | 94.62 |
| | Diff | 24.99 | 73.30 | 24.71 | 73.02 | 14.02 | 76.71 | 7.82 | 77.54 | 27.14 | 72.99 | 13.33 | 76.95 | 18.67 | 75.09 | 17.40 | 75.44 |
| LogitNorm[70] | Mean | 17.2 | 97.16 | 33.37 | 93.32 | 19.44 | 96.72 | 2.53 | 99.38 | 33.22 | 93.32 | 17.67 | 97 | 20.57 | 96.18 | 22.14 | 95.69 |
| | Std. | ± 7.07 | ± 0.79 | ± 2.05 | ± 0.42 | ± 7.97 | ± 1.25 | ± 0.62 | ± 0.43 | ± 1.68 | ± 0.34 | ± 7.72 | ± 1.14 | ± 4.52 | ± 0.68 | ± 3.47 | ± 0.64 |
| | Reported | 8.03 | 98.47 | 28.64 | 94.28 | 12.28 | 97.73 | 2.37 | 99.42 | 31.64 | 93.66 | 10.93 | 97.87 | 15.65 | 96.91 | 16.92 | 96.65 |
| | Diff | 9.17 | 1.31 | 4.73 | 0.96 | 7.16 | 1.01 | 0.16 | 0.04 | 1.58 | 0.34 | 6.74 | 0.87 | 4.92 | 0.76 | 4.33 | 0.70 |
| KNN[66] | Mean | 37.66 | 94.60 | 42.02 | 93.57 | 35.61 | 94.49 | 28.63 | 89.80 | 46.52 | 91.08 | 32.03 | 95.19 | 37.08 | 93.12 | 38.20 | 92.24 |
| | Std. | ± 8.33 | ± 1.08 | ± 2.70 | ± 0.52 | ± 3.04 | ± 0.75 | ± 2.42 | ± 19.13 | ± 1.13 | ± 0.24 | ± 2.74 | ± 0.61 | ± 3.4 | ± 3.73 | ± 3.92 | ± 4.34 |
| | Reported | 24.53 | 95.96 | 27.57 | 94.71 | 25.55 | 95.26 | 25.29 | 95.69 | 50.90 | 89.14 | | | 30.77 | 94.15 | 32.02 | 93.79 |
| | Diff | 13.13 | 1.36 | 14.45 | 1.14 | 10.06 | 0.77 | 3.34 | 5.89 | 4.38 | 1.94 | | | 6.31 | 1.03 | 4.94 | 0.96 |
| ECH ($-E(\mathbf{x})$) | Mean | 5.51 | 99.14 | 22.87 | 95.9 | 4.32 | 99.22 | 10.95 | 98.58 | 32.89 | 94.57 | 4.35 | 99.2 | 13.48 | 97.77 | 17.76 | 97.07 |
| | Std. | ± 1.16 | ± 0.14 | ± 2.51 | ± 0.36 | ± 1.65 | ± 0.17 | ± 4.58 | ± 0.44 | ± 4.29 | ± 1.03 | ± 1.39 | ± 0.16 | ± 2.6 | ± 0.38 | ± 2.41 | ± 0.42 |
| ECH ($\max'_y(p(y'|\mathbf{x}))$) | Mean | 7.55 | 98.92 | 25.81 | 95.73 | 6.55 | 98.87 | 14.49 | 97.65 | 37.77 | 94.2 | 6.66 | 98.81 | 16.47 | 97.36 | 21.16 | 96.61 |
| | Std. | ± 2.1 | ± 0.26 | ± 2.52 | ± 0.32 | ± 2.17 | ± 0.22 | ± 6.16 | ± 1.54 | ± 4.16 | ± 0.89 | ± 2.19 | ± 0.25 | ± 3.22 | ± 0.58 | ± 3.00 | ± 0.44 |

Table 15: OOD detection results (ID: CIFAR-10)

| | | acc | CIFAR-100 FPR | AUROC | AUPR | N FPR | AUROC | AUPR | U FPR | AUROC | AUPR | OODom FPR | AUROC | AUPR | Constant FPR | AUROC | AUPR | AVG FPR | AUROC | AUPR |
|---|---|---|---|---|---|---|---|---|---|---|---|---|---|---|---|---|---|---|---|---|
| $-E(\mathbf{x})$ | LinH | 97.11 | 48 | 86.41 | 81.92 | 13.62 | 98.12 | 98.78 | 27.42 | 93.29 | 94.27 | 28.5 | 96.68 | 98.02 | 0.25 | 99.9 | 99.925 | 19.63 | 95.72 | 94.58 |
| | | ± 0.86 | ± 1.06 | ± 0.31 | ± 0.76 | ± 30.86 | ± 2.55 | ± 1.5 | ± 33.9 | ± 10.1 | ± 9 | ± 45.95 | ± 3.72 | ± 2.36 | ± 0.79 | ± 0.13 | ± 0.1 | ± 27.88 | ± 8.88 | ± 8.29 |
| | ECH | **99.09** | **37.41** | **93.94** | **93.75** | **0.11** | **99.81** | **99.85** | **3.05** | **99.47** | **99.61** | **0.01** | **99.72** | **99.84** | **0** | **99.96** | **99.979** | **6.76** | **98.81** | **98.61** |
| | | ± 0.78 | ± 3.77 | ± 2.69 | ± 4.37 | ± 0.17 | ± 0.14 | ± 0.12 | ± 7.66 | ± 0.81 | ± 0.6 | ± 0.03 | ± 0.78 | ± 0.44 | ± 0 | ± 0.02 | ± 0.01 | ± 1.97 | ± 0.74 | ± 0.65 |
| $\max_y p(y|\mathbf{x})$ | LinH | 96.04 | 56.76 | 86.96 | 83.93 | 20.69 | 97 | 98.09 | 52.31 | 92.5 | 94.59 | 20.12 | 97.2 | 98.42 | 0.63 | 99.19 | 99.465 | 25.09 | 95.39 | 94.9 |
| | | ± 0.69 | ± 1.76 | ± 0.29 | ± 0.76 | ± 24.53 | ± 2.2 | ± 1.33 | ± 32.47 | ± 6.6 | ± 5.38 | ± 34.54 | ± 2.42 | ± 1.33 | ± 1.58 | ± 0.76 | ± 0.5 | ± 23.28 | ± 7.05 | ± 6.48 |
| | ECH | **98.71** | **43.62** | **93.37** | **93.53** | **0.4** | **99.59** | **99.66** | **5.32** | **99.17** | **99.4** | **3.38** | **99.53** | **99.75** | **0** | **99.94** | **99.958** | **8.79** | **98.59** | **98.46** |
| | | ± 0.98 | ± 4.44 | ± 2.26 | ± 3.57 | ± 0.64 | ± 0.39 | ± 0.3 | ± 10.86 | ± 1.17 | ± 0.8 | ± 10.69 | ± 1.49 | ± 0.79 | ± 0 | ± 0.18 | ± 0.1 | ± 7.18 | ± 2.51 | ± 1.95 |

Table 16: OOD detection results (ID: SVHN)

| | acc. | CIFAR-10[38] FPR | AUROC | Texture[6] FPR | AUROC | iSUN[71] FPR | AUROC | LSUN[74] FPR | AUROC | Places365[79] FPR | AUROC | LSUN_R[74] FPR | AUROC |
|---|---|---|---|---|---|---|---|---|---|---|---|---|---|
| LinH | 97.33 | 15.42 | 95.87 | 17.04 | 95.97 | 15.5 | 95.98 | 26.2 | **97.71** | 16.52 | 95.49 | 16.48 | 95.72 |
| | ± 0.1 | ± 1.5 | ± 0.3 | ± 0.3 | ± 2.6 | ± 0.6 | ± 2.9 | ± 0.8 | ± 5.9 | ± 2.5 | ± 1.1 | ± 0.4 | ± 3.1 |
| CosH | 97.46 | 27.65 | 86.87 | 26.53 | 85.51 | 21.15 | 89.17 | 37.73 | 77.57 | 26.38 | 86.57 | 22.48 | 88.73 |
| | ± 0.03 | ± 4.3 | ± 2.2 | ± 3.7 | ± 2.3 | ± 4.1 | ± 2.3 | ± 4.5 | ± 4.2 | ± 3.9 | ± 2.3 | ± 3.1 | ± 1.9 |
| GPH | 97.35 | 29.40 | 92.38 | 28.18 | 95.03 | 32.30 | 88.94 | 40.18 | 96.63 | 28.18 | 96.05 | 23.93 | 98.45 |
| | ± 0.12 | ± 4.7 | ± 1.3 | ± 4.1 | ± 2.4 | ± 3.6 | ± 2.5 | ± 1.5 | ± 2.1 | ± 4.4 | ± 1.4 | ± 0.6 | ± 2.1 |
| ECH | 97.29 | **0.84** | **99.54** | **2.86** | **99.15** | **0.32** | **99.68** | **15.8** | 95.15 | **0.92** | **99.53** | **0.584** | **99.62** |
| | ± 0.08 | ± 0.4 | ± 0.1 | ± 0.04 | ± 1.5 | ± 0.4 | ± 0.2 | ± 0.04 | ± 10.7 | ± 4.2 | ± 0.3 | ± 0.03 | ± 0.3 |

| | CIFAR-100[38] FPR | AUROC | N FPR | AUROC | U FPR | AUROC | OODomain FPR | AUROC | Constant FPR | AUROC | AVG FPR | AUROC |
|---|---|---|---|---|---|---|---|---|---|---|---|---|
| LinH | 16.58 | 95.53 | 10.12 | 97.87 | 17.12 | 96.64 | 2.74 | 98.99 | 14.84 | 98.21 | 15.32 | 96.73 |
| | ± 1.3 | ± 0.2 | ± 7.8 | ± 1.2 | ± 13.4 | ± 2.2 | ± 5.3 | ± 1.2 | ± 33.1 | ± 3.5 | ± 6.1 | ± 2.2 |
| CosH | 28.90 | 86.27 | 7.55 | 96.11 | 12.55 | 93.46 | 4.65 | 97.84 | 0.08 | 99.91 | 19.60 | 89.82 |
| | ± 4.2 | ± 2.2 | ± 4.1 | ± 2.1 | ± 7.1 | ± 3.8 | ± 4.6 | ± 2.0 | ± 0.2 | ± 0.1 | ± 4.0 | ± 2.3 |
| GPH | 30.60 | 95.76 | 13.70 | 99.38 | 8.80 | 97.66 | 1.65 | 98.04 | 0.13 | 99.94 | 21.64 | 95.93 |
| | ± 0.9 | ± 1.2 | ± 2.4 | ± 1.1 | ± 3.4 | ± 2.1 | ± 3.7 | ± 1.0 | ± 0.3 | ± 0.1 | ± 2.7 | ± 1.5 |
| ECH | **1.56** | **99.42** | **0.04** | **99.85** | **0.08** | **99.81** | **0** | **99.93** | **0** | **99.94** | **2.09** | **99.24** |
| | ± 0.7 | ± 0.1 | ± 0.2 | ± 0.3 | ± 0.1 | ± 0.3 | ± 0 | ± 0.1 | ± 0 | ± 0 | ± 0.5 | ± 1.2 |

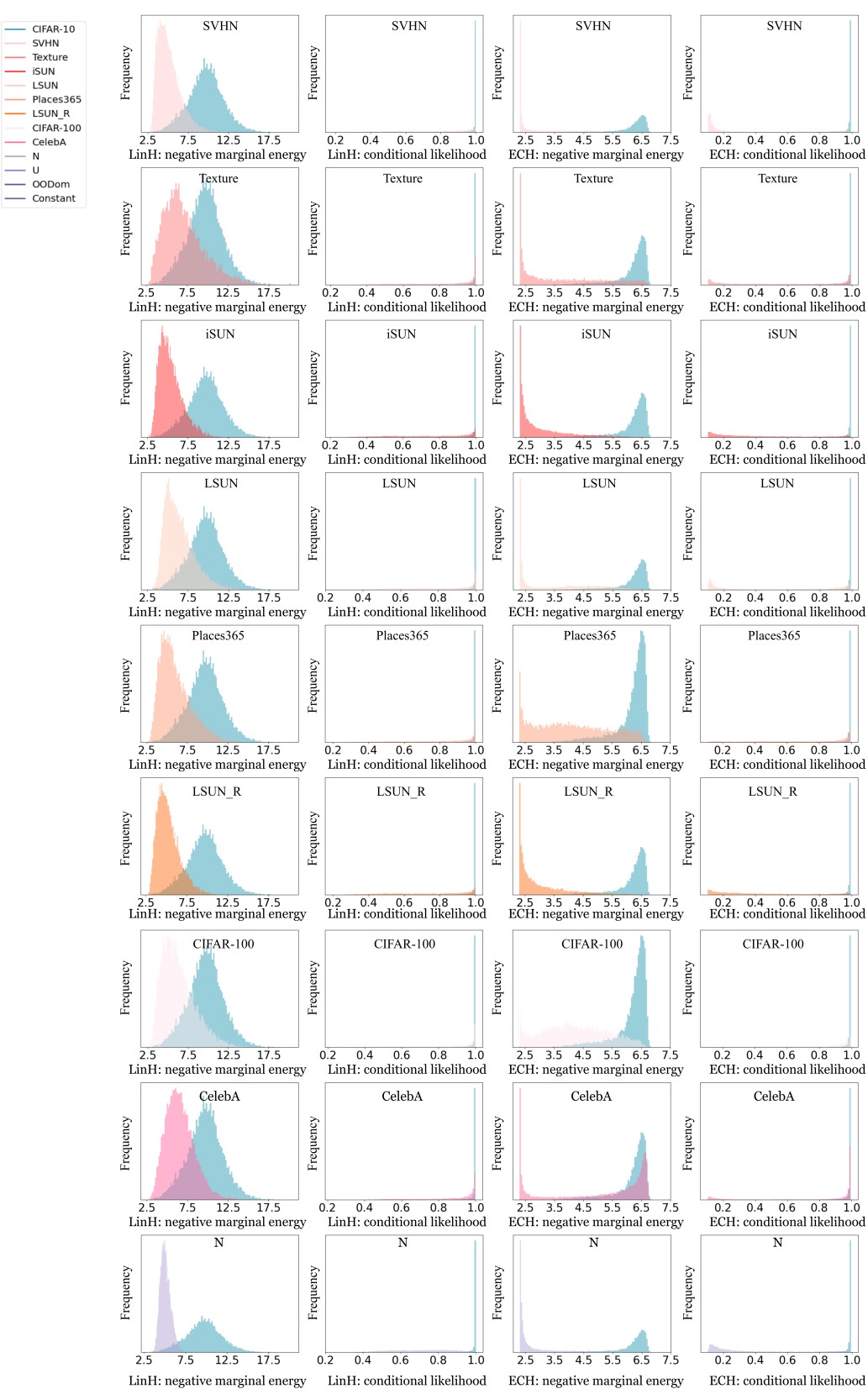

Figure 16: Histograms illustrating the distribution of conditional likelihood and marginal energies for ID (CIFAR-10) and OOD samples across different heads (1).

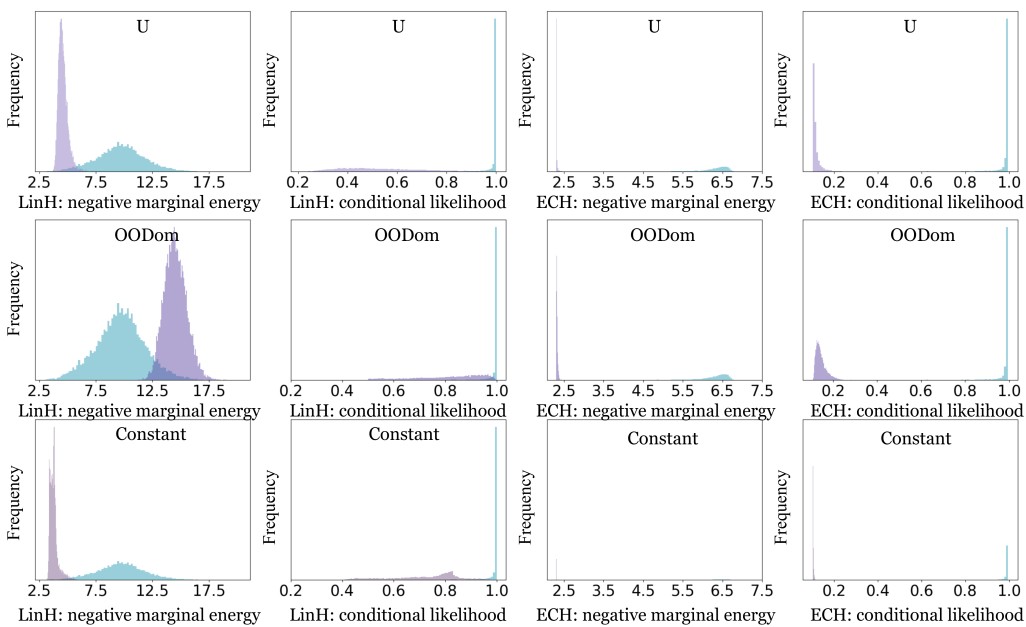

Figure 17: Histograms illustrating the distribution of conditional likelihood and marginal energies for ID (CIFAR-10) and OOD samples across different heads (2).

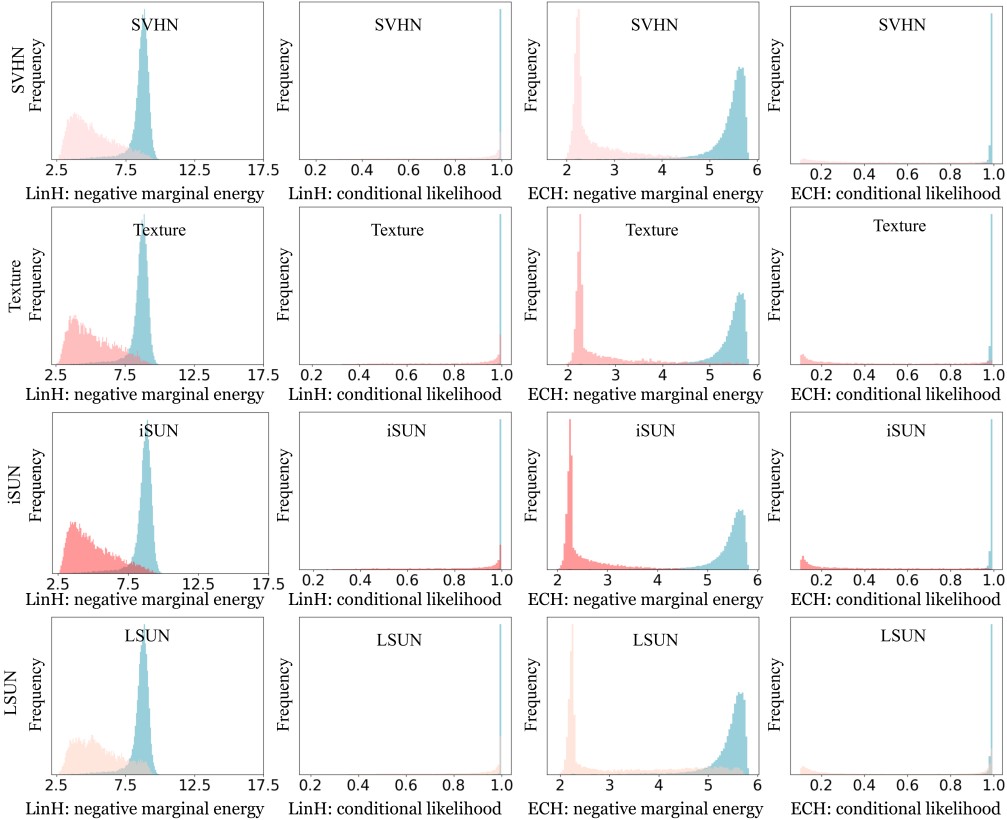

Figure 18: Histograms illustrating the distribution of conditional likelihood and marginal energies for ID (SVHN) and OOD samples across different heads (1).

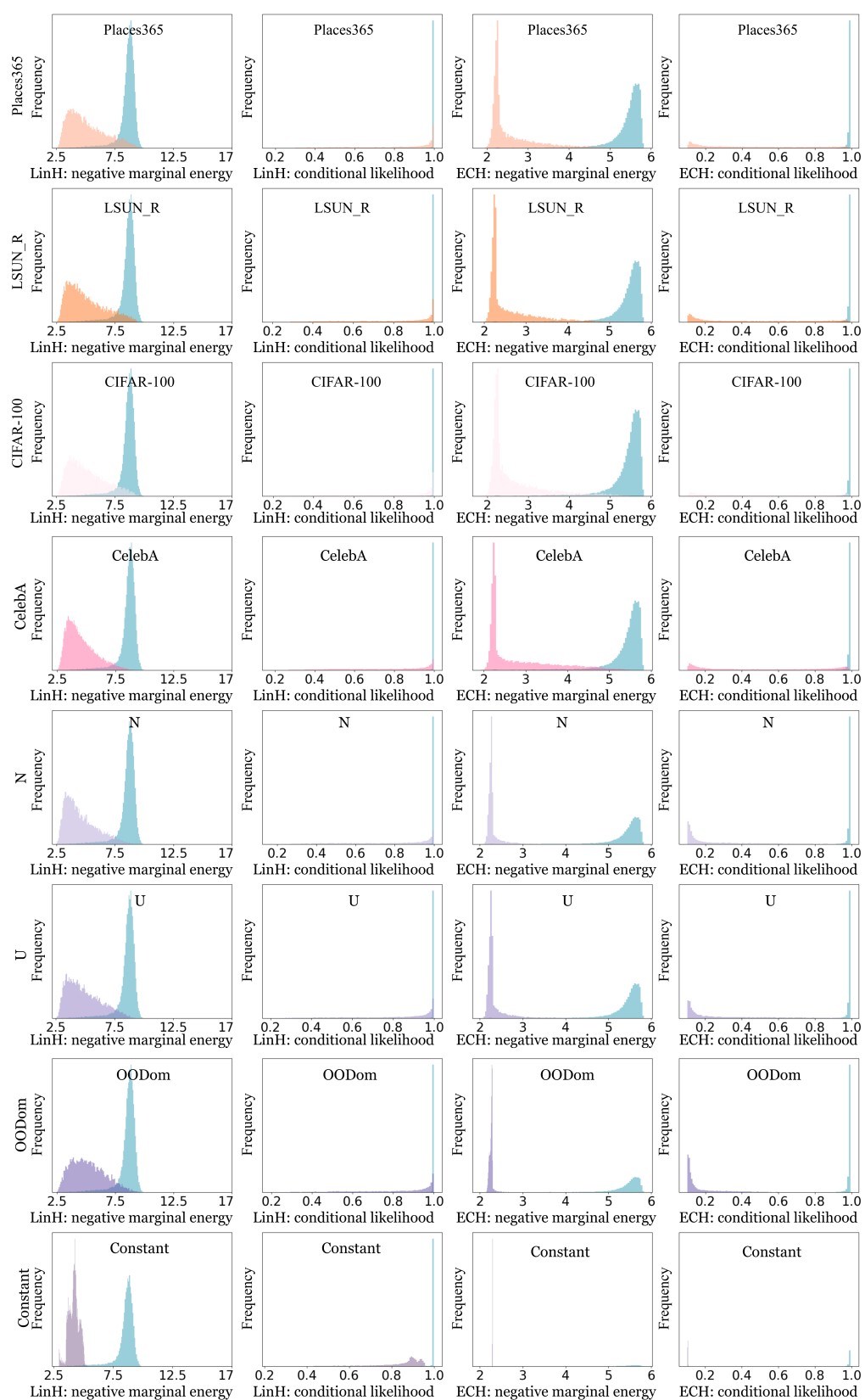

Figure 19: Histograms illustrating the distribution of conditional likelihood and marginal energies for ID (SVHN) and OOD samples across different heads (2).

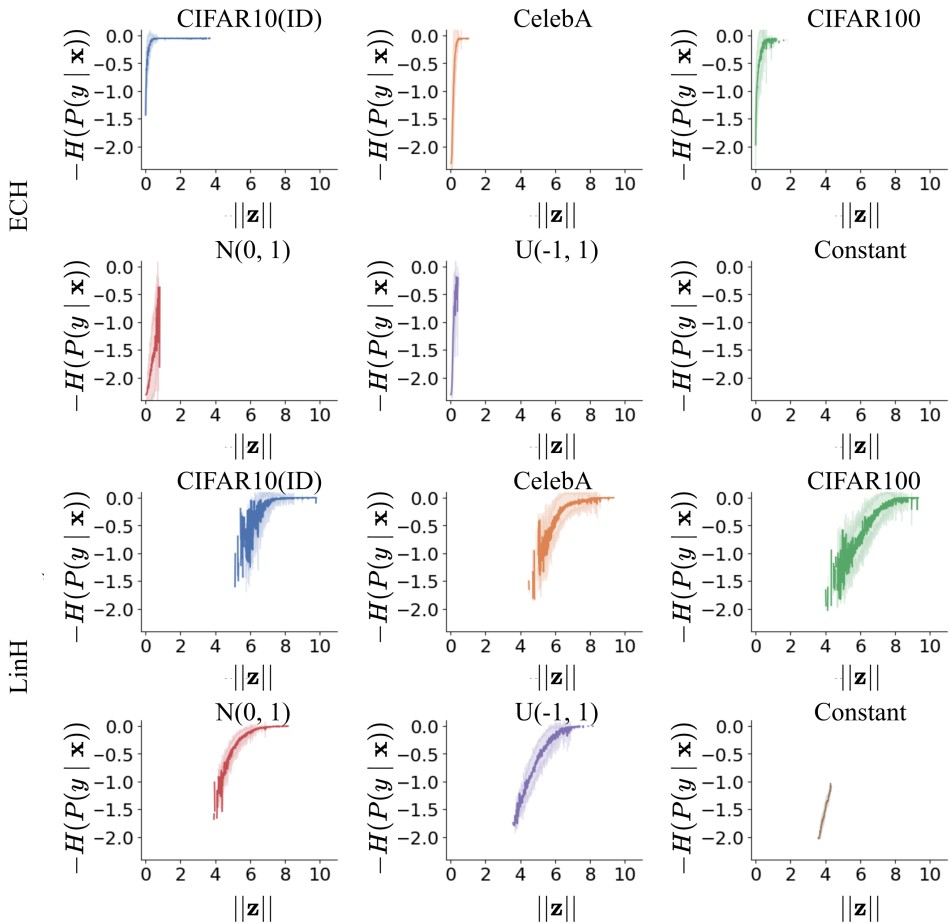

Figure 20: Influence of latent norm on $H(P(Y|\mathbf{x}))$ for each dataset

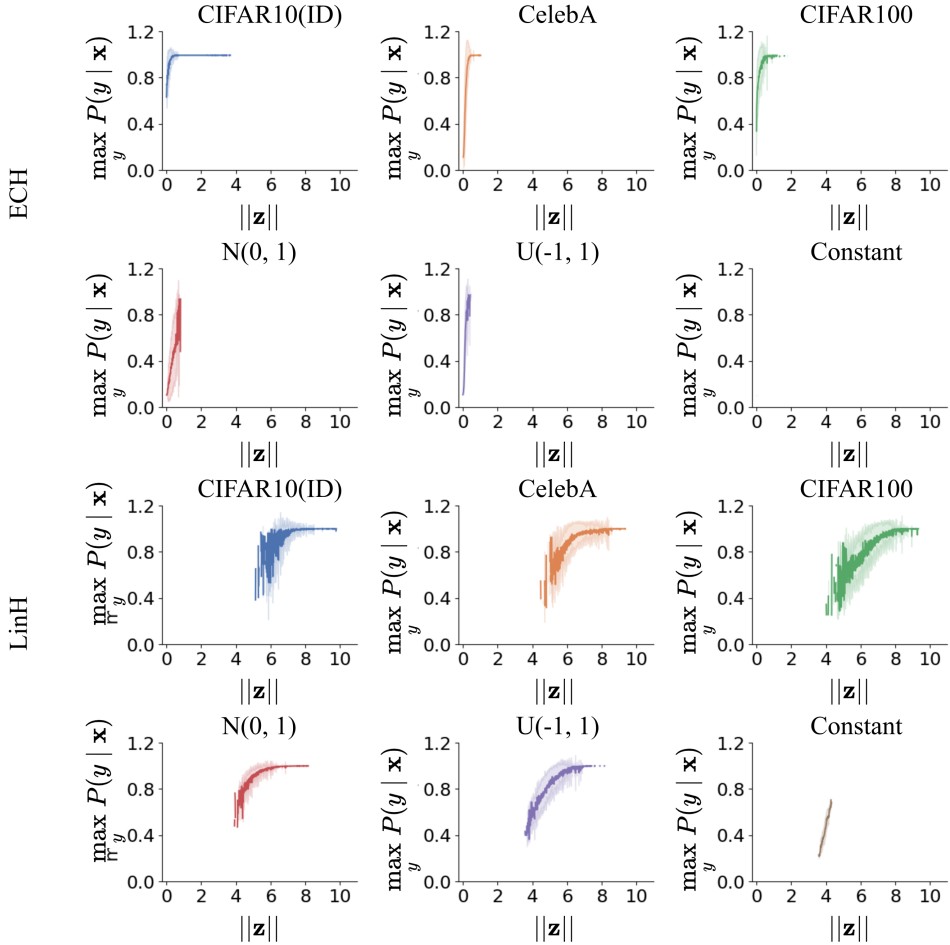

Figure 21: Influence of latent norm on $\max_y P(y|\mathbf{x})$ for each dataset

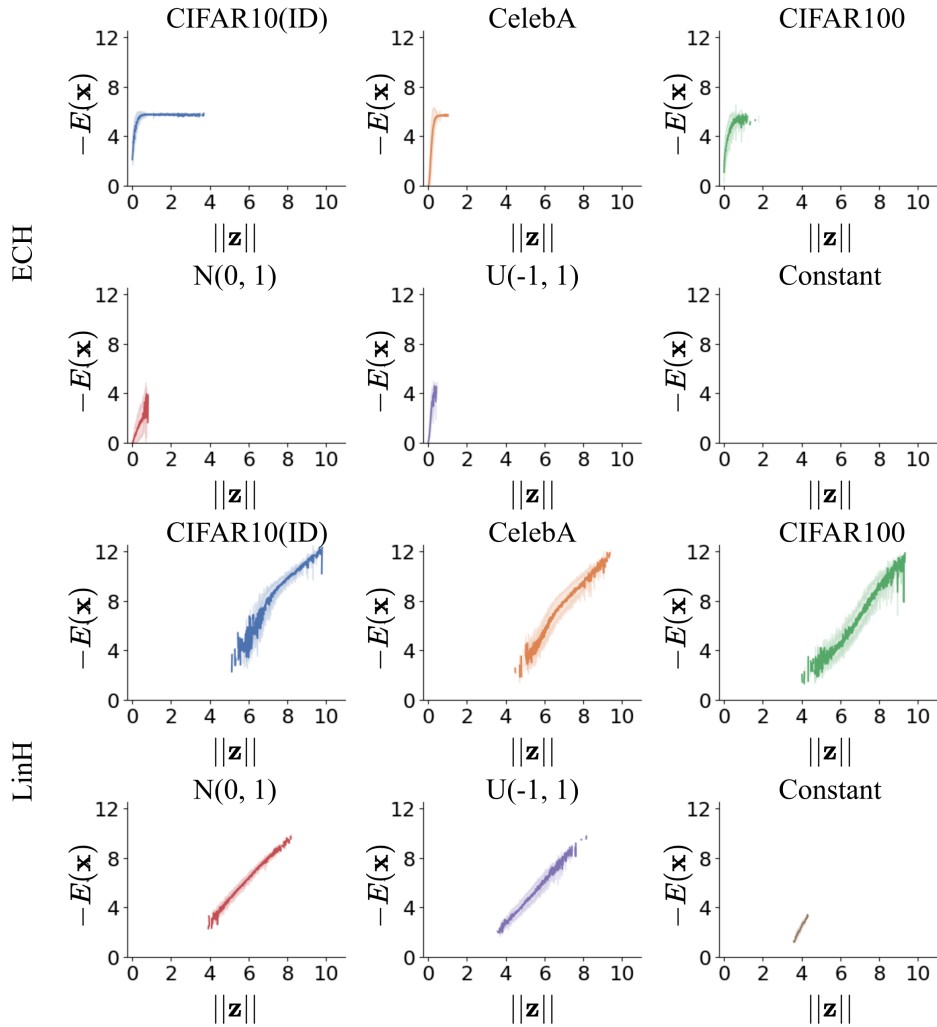

Figure 22: Influence of latent norm on $E(\mathbf{x})$ for each dataset

