# OpenReview forum: "Energy Calibration Head: A Plug-In Neural Network Head with Human-like Uncertainty"
_ICLR.cc/2024/Conference — ICLR 2024 Conference Withdrawn Submission_

### Official Review · Reviewer_BC71 · 2023-10-31

**Soundness:** 2 fair
**Presentation:** 3 good
**Contribution:** 3 good
**Rating:** 3
**Confidence:** 3

**Summary:**

Machine learning models often become overconfident on out-of-distribution (OOD) samples, unlike humans who can identify uncertainty and this study investigates the connection between marginal density and conditional uncertainty to understand the overconfidence issue.
Authors performed a human behavioral study where they collected human classification data on noisy images to show accuracy drops and uncertainty in labels and found a link between marginal probability and conditional uncertainty in human decisions.
Authors related classifiers to energy-based models and showed latent norm and biases can disrupt the connection between conditional uncertainty and marginal energy. They show that high marginal energy and uncertainty occur in OOD samples and samples with different marginal energies can have the same uncertainty level.
Authors related classifiers to energy-based models and showed latent norm (output of the penultimate layer) shows that the norm of the latent vector z directly influences both the uncertainty and marginal energy of the model and the influence on uncertainty is closely tied to the effect of biases. Finally they show that the relationship between marginal energies and conditional uncertainty in models does not match that of humans and is not strongly coupled.
Authors propose an Energy Calibration Head (ECH) plug-in layer to recalibrate uncertainty and marginal energy. ECH is intended to correct the mismatch between uncertainty and marginal energy. This ECH module calculates energy as the product of three parts: cosine similarity, a scaling factor from the metacognition module, and a scaling factor from the Kernel. The  meta-cognition MLP is used to predict marginal energy from maximum log-likelihood and expected logits. The kernel module outputs a scaling factor that converges to zero for sufficiently faraway samples and helps identifying OOD data.
Authors provide empirical analysis of the results based on the CIFAR-10 dataset and compare it with few other heads of the outline in the appendix. Authors show that their head outperforms other benchmarks they have selected for ID calibration. They also show their head leads to lower certainty for OOD samples following the behavior of humans. Finally they test their model for adversarial and OOD anomaly detection.

**Strengths:**

- The authors establish an important link between uncertainty in human decision-making and marginal probability through behavioral experiments. This provides a strong motivation for designing machine learning models that better emulate human metacognition.
- The theoretical analysis relating generic classifiers to energy-based models offers valuable insights into the connection between conditional uncertainty and marginal energy. It identifies factors that can lead to overconfident decisions.
- The proposed method provides a solution to calibrate uncertainty by resolving the inconsistency between uncertainty and marginal energy. Key advantages are its plug-in nature, no requirement for additional training(over normal ID training).
- The work makes important contributions towards achieving human-like uncertainty estimates in neural networks. This can enable more reliable AI systems and effective human-AI cooperation.

**Weaknesses:**

- Empirical evaluation is very superficial: authors only rely on a single image dataset CIFAR-10. This is a good start but more benchmarks would round up the evaluation and ensure the method really works well in practice. I would like to see authors provide results on CIFAR-100 and a larger dataset like ImageNet (or ImageNet-tiny).
- The method is not well compared to other calibration benchmarks.
---- In order to make an assessment on the quality of the results we would need to see comparisons to other calibration methods such as those in: https://arxiv.org/abs/2002.09437, https://arxiv.org/abs/1706.04599, https://proceedings.mlr.press/v80/kumar18a/kumar18a.pdf, https://arxiv.org/abs/1906.02629, https://arxiv.org/abs/1905.11001.
---- There is limited ablation analysis to thoroughly validate the impact of the different components of ECH. More controlled experiments could isolate the effects.
---- The adaptability and plug-in nature of ECH is claimed but not demonstrated extensively. More examples of integrating ECH into various models could be shown.
- The sample size for the human behavioral experiments is not massive (250 participants); I believe it is acceptable but a larger sample could have provided more robust results to establish the link between marginal probability and conditional uncertainty.

**Questions:**

- How do the calibration results of ECH compare to other state-of-the-art calibration methods like temperature scaling, vector scaling, ensembles, etc.?
- What ablation studies could be done to better isolate the impact of the different components of ECH, such as the use of the confidence model versus the ensemble approach?
- how does this method perform on other datasets such as ImageNet?

---

### Official Review · Reviewer_Gi7P · 2023-11-01

**Soundness:** 2 fair
**Presentation:** 2 fair
**Contribution:** 2 fair
**Rating:** 5
**Confidence:** 2

**Summary:**

This paper focuses on the overconfidence problem of machine learning models when encountering unseen samples.  To alleviate this problem, the authors first establish the connection between marginal probability and conditional uncertainty through human behavioral experiments on classifying noisy images. Next, theoretical analysis reveals that uncertainty and marginal energy are loosely related and significantly affected by the latent vector norm. Based on these findings, the authors propose a novel plug-in type layer, energy calibration head (ECH). Experiments on uncertainty calibration and anomaly detection demonstrate the effectiveness of their proposed method.

**Strengths:**

**Originality:** To the best of my knowledge, this paper is the first to leverage the relationship between marginal energy and uncertainty to analyze the overconfidence issue, so this work is novel.

**Quality:** The theoretical analysis in this paper is sound, and the empirical validation also provides evidence for the corresponding theorem. In addition, the authors also provide extensive experimental comparisons, including ID/OOD/distance sample calibration, adversarial, and OOD detection, to further verify the effectiveness of the proposed method.

**Reproducibility:** Code is provided.

**Weaknesses:**

1. **Clarity:** This paper is not easy to follow, especially the connection between Sec. 3 and Sec. 4. The authors mention that ECH is introduced by extending the insights of Sec. 3, where two scaling factors directly calibrate uncertainty and marginal energy. But to understand intuitively, the metacognitive module calibrates uncertainty by evaluating the difference between the actual and the predicted marginal energy. What is the purpose of the kernel module design, and what is its connection with Sec.3? Could the authors explain it in more detail?

2.  Although the author provides a variety of experimental verifications, the coverage is still somewhat limited, such as:

*  2.1. Experiments on uncertainty calibration and adversarial detection are limited to comparisons between ECH and LinH, CosH and GPH. Do the authors consider comparing other calibration algorithms, such as temperature scaling [1], focal loss [2], etc.?

[1] C. Guo, G. Pleiss, Y. Sun, and K. Q. Weinberger, “On calibration of modern neural networks,” in International conference on machine learning. PMLR, 2017, pp. 1321–1330.
[2] J. Mukhoti, V. Kulharia, A. Sanyal, S. Golodetz, P. Torr, and P. Dokania, “Calibrating deep neural networks using focal loss,” Advances in Neural Information Processing Systems, vol. 33, pp. 15 288–15 299, 2020.

*  2.2. The authors mention that the baseline selection criteria for OOD detection are designed to exclude methods that rely on synthetic data augmentation or different learning mechanisms, and specifically target the performance of six representative OOD benchmark datasets. This selection criterion seems a bit strange to me. Have the authors considered comparing representative works in Bayesian networks, ensemble algorithms, evidential networks, etc.? Including standard baselines would be quite helpful to provide a complete picture to the reader.

*  2.3. The ID dataset in the experiment only contains CIFAR-10. Do the authors consider validating on a larger ID dataset?

*  2.4 How is the accuracy of ECH compared with other baselines in the ID test dataset?

3. This paper lacks the content of related works.

4.  The unclear description and typos in the paper are as follows:

*  4.1. (Sec. 4 Metacognition Module) $\max_y \log P_h(y | \boldsymbol{x}) =  \log \text{softmax}(h_y(\boldsymbol{x}))$ ?

*  4.2. (Sec. 4 Kernel Module) $\boldsymbol{b}_y \in \mathbb{R}^{d_k} \rightarrow \boldsymbol{b}_y \in \mathbb{R}^{d_l}$

*  4.3. When LinH, CosH, and GPH first appear in Figure 1, it is recommended to provide their full names.

*  4.4. The third contribution mentions that ECH does not require additional training and data. I understand that ECH needs to be trained under ID training data, just without additional OOD data?

**Questions:**

1. ECH's learnable parameters appear to include $\phi(\cdot), \psi(\cdot), \boldsymbol{b}, \omega, \boldsymbol{w}$. Do the authors compare the parameter differences between ECH and LinH, CosH and GPH? What about the difference in time complexity of training?

2. The ablation experiment provided in Appendix Table 4 shows that using ECH of all modules is not the optimal result. Do the authors have further analysis? In addition, the mathematical symbols in Table 4 are inconsistent with the text. What do $\Phi_y(\boldsymbol{z}), \Psi(s_c(\boldsymbol{z}))$ stand for? It is recommended that the authors consider placing this ablation experiment in the main text.

3. The authors mention that ECH is a plug-in type head. The appendix also provides experiments under the WRN-28-10, ResNet18, and ResNet34 models, but these are all ResNet networks. If it is applied to the transform architecture, could it also achieve good results?

---

### Official Review · Reviewer_2EYy · 2023-11-08

**Soundness:** 2 fair
**Presentation:** 1 poor
**Contribution:** 1 poor
**Rating:** 3
**Confidence:** 4

**Summary:**

The paper proposes a new output layer that can be added at the top of neural network classifiers. The motivation for this architecture comes from human studies where a connection between marginal density and the label uncertainty is hinted at among the humans. The paper studies how the norm of the last layer representation affects both the marginal energy and the predicted label uncertainty. Based on this, the paper proposes an output layer structure that is based on quantifying error in terms of if the predicted label uncertainty (with some modifications) is predictive of the marginal energy or not, among other modifications. The proposed structure is shown to improve calibration, OOD detection in the experimental contribution.

**Strengths:**

1. The proposed contribution is a plug-in module that does not require big architectural modifications which is a plus point.
2. Study of the impact of the norm on uncertainty and marginal energy is useful.

**Weaknesses:**

1. The paper uses human like interpretation of uncertainty to better encode uncertainty in the machine learning classifiers. However, humans themselves have quite a complicated understanding of uncertainty themselves. To this end, I'm not sure I fully rely on the elicited confidence from the humans for the user study. I might also be misunderstanding Section 2 (see question 1 below).

2. The paper use EBM based interpretation of the regular classifiers to argue that the marginal energy is directly proportional to the marginal density. However, that does not happen by default which severely weakens the contribution of the paper (see question 2 below). However, I'm also curious to understand the reason behind the reported performance improvements (see question 3 and 4 below).



Some minor comments:
1. It looks like the paper needs a considerable amount of re-writing to increase the clarity (e.g. refer to question1 below). Acronyms (LinH, CosH, etc.) are used (in Fig 1 and Fig 2) without defining what they stand for until the end of the paper that hinder the clarity of the paper.

**Questions:**

Some questions follow:
1. Maybe I misunderstood something, but could authors clarify what participants' "population-level decision probability" and "class conditional uncertainty" as elicited in online experiments mean? Specifically, I'm not sure I understand the Figure 1a and 1b. In the text, it seems Fig 1a denotes the classification accuracy and in the fig, y-label says "human decision probability for the true label." Is it the proportion of participants who gave the true label? Or were the participants asked to also elicit their probability (prediction distribution) as stated in Section 5. Any help in understanding Figure 1 and section 2 is appreciated.

2. The paper uses energy based (EBM) interpretation of the generic classifier to define the marginal energy (logsumexp) as the quantity encoding marginal density. However, I believe one cannot argue that logsumexp natively captures the marginal density.  EBM interpretation does give nice and elegant parameterisation to quantify marginal density, but to be able to formally reason that proper training of the classifier is required (refer to section 4 of Grathwohl et al. (2020) paper). Standard classifiers are trained to optimise $p_{\theta}(y\vert x)$. However, to exploit EBM interpretation, one needs to also optimise $p_{\theta}(x)$. Without considering these things, the arguments in the paper are weakens. But happy to hear more from the authors in this matter.

3. Are there any ablations studies which shows how much of the OOD calibration / detection performance is coming from various components of the proposed method. I mean if I completely remove the said meta-coginition part, how much the performance diminishes. The proposed ECH also utilises distance based kernel module, I suspect that alone is contributing a lot for the OOD performance.

4. Section 5. Empirical analysis of uncertainty calibration. Could the calibration properties of the proposed method be checked for other datasets, or multiple runs?

5. How is the three-layer perceptron in the said meta-coginition module trained. Is it trained end to end with the regular classifier?

---

### Official Review · Reviewer_CexK · 2023-11-10

**Soundness:** 3 good
**Presentation:** 3 good
**Contribution:** 3 good
**Rating:** 5
**Confidence:** 3

**Summary:**

Paper studies uncertainty calibration of neural networks particularly in out-of-distribution settings and proposes a replacement for the standard classifier head in neural networks. Paper first shows the reason for overconfidence in NNs (large norms of latent representation for OOD samples) and removes the effect of these latent norms. Additionally it proposes two modules (within the last layer head) that scale down the output depending on the distance between the ID and OOD samples. Experiments show benefits in ID/OOD uncertainty calibration, adversarial and anomaly detection.

**Strengths:**

1. Paper studies an important uncertainty calibration challenge for current neural networks particularly out-of-distribution.
2. Paper proposes a plug-in classifier head that can potentially be used with any neural network architecture for improved OOD calibration.
3. Study of why neural networks are overconfident with standard classifier head (Figure 2, Thms 1&2) is interesting.
4. Empirical results show better calibration in both ID and OOD cases (including adversarial examples).

**Weaknesses:**

1. It is not clear why the proposed approach will output calibrated uncertainty in the original definition of calibration (even intuitively).
2. Paper does not provide any theoretical guarantee for calibration. I believe OOD data could be constructed such that proposed method is miscalibrated.
3. Experiments could be strengthened with naive baselines for uncertainty (Bayesian methods, Deep Ensembles) and OOD data that are similar to ID but collected under different environments (e.g., OOD benchmarks like WILDS dataset).

**Questions:**

1. Questions regarding the human behavior study in Section 2:
    1. Naming the samples ID and OOD seems a little strange (i.e., what is ID and OOD for humans?)
    2. OOD samples are constructed only with noise; why is noisy data OOD? For example, if humans are shown MNIST (ID)->SVHN (OOD) etc., these observations will possibly not hold.
    3. Does this study show humans are calibrated (i.e., probability given from humans matches their accuracy)?
    4. Finally, it would be helpful to the reader if marginal density w.r.t. $\lambda$ is clearly written/shown.
2. Please add a few lines concluding Figure 2 in the caption. I understand the conclusions from the plots for LinH (how quantities depend on ||z||, etc.) but I am unable to compare between the plots for LinH with plots for ECH. Do these plots show ECH is better? If so, how?
3. In the kernel module, what stops $\psi$ to learn to be a constant zero function? Does this module rely on $\psi$ not being able to learn zero function OOD?
4. How important is the metacognition module? If I am interpreting the experiments in Appx G correctly, approach without this module performs best in most tasks.
5. I am not very familiar with OOD detection approaches and the kernel module seems to rely on this. Could the authors please discuss how good these distance-based detection approaches are, especially for high-dimensional data? Are these distances meaningful: for example, if MNIST is ID, do we get lower distances for a different digit dataset vs an image dataset.
6. Is the proposed approach guaranteed to be always calibrated (in the original definition of calibration or calibration curves in Figure 4)? I am not sure if this is true and one may be able to construct OOD data where the method is not calibrated. For example, a miscalibrated model may be constructed by changing P(X) without changing its support.
7. For adversarial detection, were the adversarial examples generated for the proposed model? It is surprising that adversarial attacks are unable to find close-by examples that give incorrect probabilities.
8. Table 2 does contain quite a few baselines. However, I think it would help to show that simple ways of obtaining uncertainty fail (for example, different Bayesian methods, Deep Ensembles (https://arxiv.org/abs/1612.01474), etc).
9. Many of the results consist of OOD datasets quite different from ID dataset. It will be interesting to see if the method is able to perform well in cases when OOD dataset is similar to ID, for example, in WILDS dataset (https://wilds.stanford.edu/datasets/), similar images are collected under different environments (each with different distribution).

---

### Author Response · Authors · 2023-11-22

We express gratitude for the time and effort dedicated to reviewing our work.
We recognize the inadequacy of our preparation and, consequently, have decided to withdraw our submitted work.
Our primary research focus is on addressing the issue of overconfidence concerning Out-of-Distribution (OOD) samples.
In pursuit of this objective, our revision plan is to define a novel problem concerning OOD Calibration, distinct from model calibration, and underscore the approaches of existing methods in model calibration or OOD detection to tackle this issue.
Moreover, we plan to supplement various additional experiments and theoretical claims suggested by the reviewers.
We highly value your diligent critiques, which will serve as invaluable guidance in furthering our research.